# DAW: Exploring the Better Weighting Function for Semi-supervised Semantic Segmentation

**Rui Sun**[1*]  **Huayu Mai**[1*]  **Tianzhu Zhang**[1,2†]  **Feng Wu**[1,2]

[1]Deep Space Exploration Laboratory/School of Information Science and Technology,
University of Science and Technology of China
[2]Institute of Artificial Intelligence, Hefei Comprehensive National Science Center
`{issunrui, mai556}@mail.ustc.edu.cn, {tzzhang, fengwu}@ustc.edu.cn`

## Abstract

The critical challenge of semi-supervised semantic segmentation lies how to fully exploit a large volume of unlabeled data to improve the model's generalization performance for robust segmentation. Existing methods tend to employ certain criteria (weighting function) to select pixel-level pseudo labels. However, the trade-off exists between inaccurate yet utilized pseudo-labels, and correct yet discarded pseudo-labels in these methods when handling pseudo-labels without thoughtful consideration of the weighting function, hindering the generalization ability of the model. In this paper, we systematically analyze the trade-off in previous methods when dealing with pseudo-labels. We formally define the trade-off between inaccurate yet utilized pseudo-labels, and correct yet discarded pseudo-labels by explicitly modeling the confidence distribution of correct and inaccurate pseudo-labels, equipped with a unified weighting function. To this end, we propose Distribution-Aware Weighting (DAW) to strive to minimize the negative equivalence impact raised by the trade-off. We find an interesting fact that the optimal solution for the weighting function is a hard step function, with the jump point located at the intersection of the two confidence distributions. Besides, we devise distribution alignment to mitigate the issue of the discrepancy between the prediction distributions of labeled and unlabeled data. Extensive experimental results on multiple benchmarks including mitochondria segmentation demonstrate that DAW performs favorably against state-of-the-art methods. Code is available at `https://github.com/yuisuen/DAW`.

## 1  Introduction

Semantic segmentation is a fundamental task that has achieved conspicuous achievements credited to the recent advances in deep neural networks [1]. However, its data-driven nature makes it heavily dependent on massive pixel-level annotations, which are laborious and time-consuming to gather. To alleviate the data-hunger issue, considerable works [2–7] have turned their attention to semi-supervised semantic segmentation, which has demonstrated great potential in practical applications [8, 9]. Since only limited labeled data is accessible, how to fully exploit a large volume of unlabeled data to improve the model's generalization performance for robust segmentation is thus extremely challenging.

In previous literature, pseudo-labeling [10–12] and consistency regularization [13–15] have emerged as mainstream paradigms to leverage unlabeled data for semi-supervised segmentation. In specific,

---

[*]Equal contribution
[†]Corresponding author

37th Conference on Neural Information Processing Systems (NeurIPS 2023).

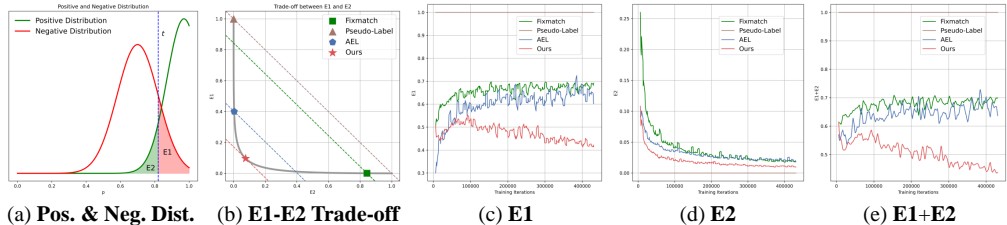

| (a) **Pos. & Neg. Dist.** | (b) **E1-E2 Trade-off** | (c) **E1** | (d) **E2** | (e) **E1+E2** |

Figure 1: Illustration of our motivation. (a) shows the trade-off between inaccurate yet utilized pseudo-labels, and correct yet discarded pseudo-labels by explicitly modeling the confidence distribution of correct and inaccurate pseudo-labels. (b) illustrates the negative equivalence impact on generalization performance raised by the trade-off. (c) (d) (e) summarize the models inevitably face a trade-off when dealing with pseudo-labels. Our method can guarantee the theoretical optimal solution by minimizing the negative impact.

the pseudo-labeling methods train the model on unlabeled samples with pseudo labels derived from the up-to-date model's own predictions. And the consistency regularization methods encourage the model to produce consistent predictions for the same sample with different perturbation views, following the smoothness assumption [16]. Recently, these two paradigms are often intertwined in the form of a teacher-student scheme [17–20, 3]. The critical idea involves updating the weights of the teacher model using the exponential moving average (EMA) of the student model, and the teacher model generates corresponding pseudo labels of the perturbed samples to instruct the learning of the student model.

Despite yielding promising results, these methods tend to employ certain criteria (referred to as *weighting function*) to select pixel-level pseudo labels, considering that the quality of the chosen pseudo-labels determines the upper bound of performance. On the one hand, naive pseudo-labeling methods such as Pseudo-Label [10] recruit all pseudo labels into training, assuming that each pseudo label is equally correct (i.e., weighting function can be regarded as a constant function). However, as training progresses, maximizing the utilization of pseudo-labels tends to lead to confirmation bias [21], which is a corollary raised by erroneous pseudo-labels (i.e., *inaccurate yet utilized pseudo-labels*). On the other hand, a series of threshold-based pseudo-labeling methods [22] such as FixMatch [18] attempt to set a high threshold (e.g., 0.95) to filter out pixel-level pseudo-labels with low confidence (i.e., the weighting function can be considered as a step function that jumps at 0.95). Although tangling the quality of pseudo-labels can alleviate noise, the strict criteria inevitably lead to the contempt of numerous unconfident yet correct pseudo-labels (i.e., *correct yet discarded pseudo-labels*), hindering the learning process. To make matters worse, the negative impact is inevitably amplified by inbuilt low-data regimes of semi-supervised segmentation, leading to sub-optimal results. As a compromise, AEL [19] ad hoc defines the weighting function as a power function, which assigns weights conditioned on the confidence of pseudo-labels, that is, convincing pseudo-labels will be allocated more weights. However, the lack of sophisticated consideration and the arbitrary control of hyperparameters (i.e., tunable power) for the weighting function inevitably compromise its capability. In a nutshell, the trade-off exists between inaccurate yet utilized pseudo-labels, and correct yet discarded pseudo-labels in these methods when handling pseudo-labels without thoughtful consideration of the weighting function, hindering the generalization ability of the model. Then, the question naturally arises: *How to explore the better weighting function to effectively alleviate the negative impact raised by the trade-off?*

In this work, we systematically analyze the trade-off in previous methods that hinder the model's learning when dealing with pseudo-labels in semi-supervised semantic segmentation. We formally define the trade-off between inaccurate yet utilized pseudo-labels, and correct yet discarded pseudo-labels by explicitly modeling the confidence distribution of correct and inaccurate pseudo-labels, equipped with a unified weighting function. In specific, two Gaussian functions excelling at the maximum entropy property are devised to fit the confidence distribution of correct (positive distribution in Figure 1 (a)) and inaccurate (negative distribution in Figure 1 (a)) pseudo-labels using maximum likelihood estimation, respectively. The parameters of the Gaussian functions are updated via the exponential moving average (EMA) in pursuit of perceiving the learning status of the model. Then the trade-off can be naturally derived by calculating the expectations of inaccurate yet utilized pseudo-labels (depicted as $E_1$ in Figure 1 (a)), and correct yet discarded pseudo-labels (displayed as

Table 1: We analyze the learning process of the mainstream methods for semi-supervised semantic segmentation systematically and uniformly abstract the criteria they adopt to select pseudo labels as *weighting function $f(p)$* conditioned on the *confidence $p$* of pseudo-labels. There are *inherent distributions $g(p)$* for positive and negative pseudo-labels(Gaussian distribution is employed as an approximation).

| Method | Pseudo-Label | Fixmatch | AEL | Ours |
|---|---|---|---|---|
| $f(p)$ | $f(p) = u(p)$ | $f(p) = u(p-t)$ | $f(p) = p^2$ | $f(p) = u(p-t^*)$ |
| $f(p) \cdot g(p)$ | *(E1)* | *(E2, E1)* | *(E2, E1)* | *(E2, E1)* |
| $E1$ | $1$ | $\beta[\Phi(\frac{1-\mu^-}{\sigma^-}) - \Phi(\frac{t-\mu^-}{\sigma^-})]$ | $(\mu^-)^2 + (\sigma^-)^2 + \mu^-(\sigma^-)^2 g^-(\frac{1}{C}) - (1+\mu^-)(\sigma^-)^2 g^-(1)$ | $\beta[\Phi(\frac{1-\mu^-}{\sigma^-}) - \Phi(\frac{t^*-\mu^-}{\sigma^-})]$ |
| $E2$ | $0$ | $1 - \alpha[\Phi(\frac{1-\mu^+}{\sigma^+}) - \Phi(\frac{t-\mu^+}{\sigma^+})]$ | $1 - (\mu^+)^2 - (\sigma^+)^2 - \mu^+(\sigma^+)^2 g^+(\frac{1}{C}) + (1-\mu^+)(\sigma^+)^2 g^+(1)$ | $1 - \alpha[\Phi(\frac{1-\mu^+}{\sigma^+}) - \Phi(\frac{t^*-\mu^+}{\sigma^+})]$ |
| $E1 + E2$ | $1 + \beta[\Phi(\frac{1-\mu^-}{\sigma^-}) - \Phi(\frac{t^*-\mu^-}{\sigma^-})] - \alpha[\Phi(\frac{1-\mu^+}{\sigma^+}) - \Phi(\frac{t^*-\mu^+}{\sigma^+})] + \int_{\frac{1}{C}}^{t^*}[g^-(p) - g^+(p)]\,dp$ | $1 + \beta[\Phi(\frac{1-\mu^-}{\sigma^-}) - \Phi(\frac{t^*-\mu^-}{\sigma^-})] - \alpha[\Phi(\frac{1-\mu^+}{\sigma^+}) - \Phi(\frac{t^*-\mu^+}{\sigma^+})] - \int_{t^*}^{t}[g^-(p) - g^+(p)]\,dp$ | $1 + \beta[\Phi(\frac{1-\mu^-}{\sigma^-}) - \Phi(\frac{t^*-\mu^-}{\sigma^-})] - \alpha[\Phi(\frac{1-\mu^+}{\sigma^+}) - \Phi(\frac{t^*-\mu^+}{\sigma^+})] + \int_{t^*}^{\frac{1}{C}} x^2[g^-(p) - g^+(p)]\,dp + \int_{t^*}^{1}(x^2-1)[g^-(p) - g^+(p)]\,dp$ | $1 + \beta[\Phi(\frac{1-\mu^-}{\sigma^-}) - \Phi(\frac{t^*-\mu^-}{\sigma^-})] - \alpha[\Phi(\frac{1-\mu^+}{\sigma^+}) - \Phi(\frac{t^*-\mu^+}{\sigma^+})]$ |
| Note | $\int_{\frac{1}{C}}^{t^*}[g^-(p) - g^+(p)]\,dp > 0$ | $-\int_{t^*}^{t}[g^-(p) - g^+(p)]\,dp \geqslant 0$ | $\int_{\frac{1}{C}}^{t^*} x^2[g^-(p) - g^+(p)]\,dp + \int_{t^*}^{1}(x^2-1)[g^-(p) - g^+(p)]\,dp > 0$ | Smaller than all of them! |

$E_2$ in Figure 1 (a)) respectively, on top of the weighting function and the corresponding confidence distribution. Now, we are prepared to propose the Distribution-Aware Weighting (*DAW*) function striving to minimize the negative equivalence impact on generalization performance raised by the trade-off, i.e., minimizing $E1 + E2$. By leveraging functional analysis, we find an interesting fact that the optimal solution for the weighting function is a hard step function, with the jump point located at the intersection of the two confidence distributions. Note that the dedicated weighting function is theoretically guaranteed by reconciling the intrinsic tension between $E1$ and $E2$ (see Figure 1 (b)), and is free of setting thresholds manually compared to previous methods. Besides, considering the imbalance issue caused by the discrepancy between the prediction distributions of labeled and unlabeled data, we propose distribution alignment to further unlock the potential of the weighting function and enjoy the synergy. In practice, the weighting function generated by DAW determines the criteria for selecting pseudo-labels to minimize the negative equivalence impact of the trade-off, minimizing $E1 + E2$ (see Figure 1 (e)), which is conducive to model training. In this way, the model improves, benefiting from the effective probe of reliable pseudo-labels. And in turn, the positive distribution will be maximally separated from the negative one, leading to a simultaneous decrease in both $E1$ and $E2$ (see Figure 1 (c) and Figure 1 (d)), which is conducive to the generation of the weighting function.

Extensive experiments on mainstream benchmarks demonstrate that our method performs favorably against state-of-the-art semi-supervised semantic segmentation methods, proving that it can better exploit unlabeled data. Besides, we further validate the robustness of DAW on the electron microscopy mitochondria segmentation task, which involves images with dense foreground objects and cluttered backgrounds, making it more challenging to discriminate the reliability of the pseudo-labels.

## 2 Distribution-Aware Weighting

In this section, we first formulate the semi-supervised semantic segmentation problem as preliminaries (Section 2.1), and then formally define the trade-off between inaccurate yet utilized pseudo-labels ($E1$), and correct yet discarded pseudo-labels ($E2$) by explicitly modeling the confidence distribution of correct and inaccurate pseudo-labels, equipped with a unified weighting function (Section 2.2).

Based on the analysis, we propose the distribution-aware weighting function (DAWF) to avoid performance degradation raised by the trade-off (Section 2.3). Finally, distribution alignment (DA) is devised to alleviate the discrepancies between the confidence distributions of labeled and unlabeled data. (Section 2.4).

## 2.1 Preliminaries

In semi-supervised semantic segmentation, given a set of labeled training images $\mathcal{D}_l = \left\{ \mathbf{x}_i^l, \mathbf{y}_i^l \right\}_{i=1}^{N_l}$ and a large amount of unlabeled images $\mathcal{D}_u = \{\mathbf{x}_i^u\}_{i=1}^{N_u}$, where $N_l$ and $N_u$ denote the number of labeled and unlabeled images, respectively, and $N_u \gg N_l$. Let $\mathbf{q}(\mathbf{x}_{ij}^*) \in \mathbb{R}^C$ denotes the prediction of the $j$-th pixel in the $i$-th labeled (or unlabeled) image, and $* \in \{l, u\}$, $C$ is the number of categories. Then the supervised loss $\mathcal{L}_s$ can be formulated as,

$$\mathcal{L}_s = \frac{1}{N_l} \sum_{i=1}^{N_l} \frac{1}{WH} \sum_{j=1}^{WH} \ell_{ce} \left( \mathbf{y}_{ij}^l, \mathbf{q}(\mathbf{x}_{ij}^l) \right), \tag{1}$$

where $W$ and $H$ represent the width and height of the input image, $\ell_{ce}$ denotes the standard pixel-wise cross-entropy loss, and $\mathbf{y}_{ij}^l$ denotes the ground-truth label from $\mathcal{D}_l$. Considering most methods [22, 18, 19, 10, 3, 23] tend to employ certain criteria (weighting function) to attempt to select reliable pseudo labels, we formulate the unsupervised loss $\mathcal{L}_u$ as weighted cross-entropy for the convenience of introducing the weighting function $f(p_{ij})$,

$$\mathcal{L}_u = \frac{1}{N_u} \sum_{i=1}^{N_u} \frac{1}{WH} \sum_{j=1}^{WH} f(p_{ij}) \cdot \ell_{ce} \left( \hat{\mathbf{y}}_{ij}^u, \mathbf{q}(\mathcal{A}^s(\mathcal{A}^w(\mathbf{x}_{ij}^u))) \right), \tag{2}$$

where $\mathcal{A}^w/\mathcal{A}^s$ denotes weak/strong perturbation to encourage the model to produce consistent predictions, $\mathbf{y}_{ij}^u$ denotes $\mathbf{q}(\mathcal{A}^w(\mathbf{x}_{ij}^u))$, i.e., prediction under the weak perturbation view. And $\hat{\mathbf{y}}_{ij}^u = \operatorname{argmax}(\mathbf{y}_{ij}^u)$ is the one-hot pseudo-label, $f(p_{ij})$ is a weighting function conditioned on $p_{ij}$, and $p_{ij} = \max(\mathbf{y}_{ij}^u)$, denotes the maximum confidence of the prediction. Then we define the overall loss function as $\mathcal{L} = \mathcal{L}_s + \mathcal{L}_u$.

## 2.2 E1-E2 Trade-off from Unified Weighting Function

We formally define the trade-off between inaccurate yet utilized pseudo-labels, and correct yet discarded pseudo-labels by explicitly modeling the confidence distribution of correct and inaccurate pseudo-labels, equipped with a unified weighting function. We instantiate the different biases inherent in the trade-off of previous methods and reveal their tight connection with the capability of the model. We start by defining the pseudo-label confidence distribution.

Orthogonal to other previous models, we assume that the confidence distribution of correct ($g^+(p)$, positive distribution) and inaccurate ($g^-(p)$, negative distribution) pseudo-labels follows a truncated Gaussian distribution with mean $\mu^+/\mu^-$ and standard deviation $\sigma^+/\sigma^-$, formulated as,

$$g^+(p) = \begin{cases} \frac{\alpha}{\sqrt{2\pi}\sigma^+} \exp\left[-\frac{(p-\mu^+)^2}{2(\sigma^+)^2}\right], & \frac{1}{C} \leqslant p \leqslant 1 \\ 0, & \text{otherwise} \end{cases}, \tag{3}$$

where $1/\alpha = \Phi\left(\frac{1-\mu^+}{\sigma^+}\right) - \Phi\left(\frac{1/C-\mu^+}{\sigma^+}\right)$ denotes the normalization factor, $\Phi$ is the cumulative distribution function of the standard normal distribution. Note that $p = \max(\mathbf{q}(\mathbf{x}_{ij}^l))$, denotes the maximum confidence of the prediction from labeled data, so it must meet the condition of $p \geqslant \frac{1}{C}$. Similarly,

$$g^-(p) = \begin{cases} \frac{\beta}{\sqrt{2\pi}\sigma^-} \exp\left[-\frac{(p-\mu^-)^2}{2(\sigma^-)^2}\right], & \frac{1}{C} \leqslant p \leqslant 1 \\ 0, & \text{otherwise} \end{cases}, \tag{4}$$

where $1/\beta = \Phi\left(\frac{1-\mu^-}{\sigma^-}\right) - \Phi\left(\frac{1/C-\mu^-}{\sigma^-}\right)$. The reason we choose Gaussian is its valuable maximum entropy property, please refer to the supplementary material for more details. Then we estimate

the mean and standard deviation of the positive $g^+(p)$ and negative distribution $g^-(p)$, respectively, resorting to maximum likelihood estimation,

$$\hat{\mu}^+ = \frac{1}{N_l} \sum_{i=1}^{N_l} \frac{1}{N_i^+} \sum_{j=1}^{N_i^+} p_{ij}^+, \qquad (\hat{\sigma}^+)^2 = \frac{1}{N_l} \sum_{i=1}^{N_l} \frac{1}{N_i^+} \sum_{j=1}^{N_i^+} (p_{ij}^+ - \hat{\mu}^+)^2, \qquad (5)$$

where $p_{ij}^+$ denotes the prediction confidence, where $\mathrm{argmax}(\mathbf{q}(\mathbf{x}_{ij}^l))$ on the labeled data equals the ground truth $\mathbf{y}_{ij}^l$, and $N_i^+$ is the number of $p_{ij}^+$ in the $i$-th image. For the negative distribution $g^-(p)$, the parameters are evaluated in the same way, except that the predictions involved in the calculation are not equal to the ground truth. Note that we only consider predictions from labeled data to evaluate the Gaussian distribution equipped with ground truth, rather than estimating biases raised from unlabeled data with noisy pseudo-labels. Then the parameters of the Gaussian functions are updated via the exponential moving average (EMA) in pursuit of perceiving the learning status of the model in a dynamic manner,

$$\hat{\mu}_t^+ = m\hat{\mu}_{t-1}^+ + (1-m)\hat{\mu}^+, \qquad (\hat{\sigma}_t^+)^2 = m(\hat{\sigma}_{t-1}^+)^2 + (1-m)\frac{\sum_i N_i}{\sum_i N_i - 1}(\hat{\sigma}^+)^2, \qquad (6)$$

where unbiased variance is adopted for EMA, $\hat{\mu}_0^+$ and $(\hat{\sigma}_0^+)^2$ are initialized as $1/C$ and $1.0$ respectively. A similar way also works for the negative distribution $g^-(p)$.

Then the trade-off can be naturally derived by calculating the expectations of inaccurate yet utilized pseudo-labels ($E1$), and correct yet discarded pseudo-labels ($E2$) respectively, on top of the weighting function $f(p)$ and the corresponding confidence distribution $g^-(p)/g^+(p)$.

**Definition 3.1** Inaccurate yet utilized pseudo-labels, $E1$.

$$E1 = \mathbb{E}_{g^-}[f(p)] = \int_{\frac{1}{C}}^1 f(p) \cdot g^-(p) dp. \qquad (7)$$

**Definition 3.2** Correct yet discarded pseudo-labels, $E2$.

$$E2 = 1 - \mathbb{E}_{g^+}[f(p)] = 1 - \int_{\frac{1}{C}}^1 f(p) \cdot g^+(p) dp. \qquad (8)$$

After formally defining the trade-off between $E1$ and $E2$, it is natural to measure the impact of negative equivalence effects (i.e., $E1+E2$), considering the trade-off between $E1$ and $E2$, where an increase in one necessitates a decrease in the other.

**Definition 3.3** Negative equivalence effect of the trade-off, $E1+E2$.

$$E1 + E2 = 1 + \int_{\frac{1}{C}}^1 f(p) \cdot \left[ g^-(p) - g^+(p) \right] dp, \qquad (9)$$

Then, we systematically analyze the trade-off in previous methods as tabulated in Table 1. For more detailed derivations, please refer to the supplementary material. (1) For example, naive pseudo-labeling methods such as Pseudo-Label [10] enroll all pseudo labels ($E2 = 0$) into training. However, as training progresses, maximizing the utilization of pseudo-labels tends to a confirmation bias raised by erroneous pseudo-labels ($E1 = 1$). (2) And for threshold-based pseudo-labeling methods such as FixMatch [18], which attempts to set a high threshold (0.95) to filter out pixel-level pseudo-labels with low confidence (*small value* of $E1$ caused by the proximity of $t = 0.95$ to 1). However, the strict criteria inevitably lead to the contempt of numerous unconfident yet correct pseudo-labels (*large value* of $E2$ caused by the proximity of $t = 0.95$ to 1). (3) As a compromise, AEL [19] ad hoc defines the weighting function as a power function, which assigns weights conditioned on confidence. That is, convincing pseudo-labels will be allocated more weight. However, the lack of sophisticated consideration and the arbitrary control of hyperparameters (tunable power) for the weighting function inevitably compromise its capability (not guaranteeing the lowest negative equivalence effect).

## 2.3 Distribution-Aware Weighting Function

Then we seek to explore a better weighting function equipped with the formal trade-off definition, aiming at minimizing the negative equivalence impact raised by the trade-off, that is, minimizing

$E1 + E2$,

$$\min_{f(p)} \quad E1 + E2 = 1 + \int_{\frac{1}{C}}^{1} f(p) \cdot [g^-(p) - g^+(p)] \, dp, \quad (10)$$
$$\text{s.t.} \quad 0 \leq f(p) \leq 1,$$

By leveraging functional analysis, we find an interesting fact that the optimal solution for the weighting function is a hard step function, with the jump point located at the intersection of the two confidence distributions, formulated as,

$$f^*(p) = \left\{ \begin{array}{ll} 1, & t^* \leqslant p \leqslant 1 \\ 0, & \text{otherwise} \end{array} \right. , \quad t^* = \left( \left( \beta_2^2 - 4\beta_1\beta_3 \right)^{\frac{1}{2}} - \beta_2 \right) / (2\beta_1) , \quad (11)$$

where $\beta_1 = (\sigma^+)^2 - (\sigma^-)^2, \beta_2 = 2[\mu^+(\sigma^-)^2 - \mu^-(\sigma^+)^2], \beta_3 = (\sigma^+\mu^-)^2 - (\sigma^-\mu^+)^2 + 2(\sigma^+\sigma^-)^2 \ln[(\alpha\sigma^-)/(\beta\sigma^+)]$ and $p = \max(\mathbf{y}_{ij}^u)$ denotes the confidence of the prediction from unlabeled data. Please refer to the supplementary material for more detailed derivations. Note that the dedicated weighting function $f^*(p)$ is theoretically guaranteed by reconciling the intrinsic tension between $E1$ and $E2$ (see Table 1) and is free of setting thresholds manually compared to previous methods.

## 2.4 Distribution Alignment

Furthermore, considering the imbalance issue caused by the discrepancy between the prediction distributions of labeled and unlabeled data, we propose distribution alignment (DA) to further unlock the potential of the distribution-aware weighting function. In specific, we define the confidence distributions from labeled data and unlabeled data as expectations $\mathbb{E}_{\mathcal{D}_l}\left[\mathbf{q}(\mathbf{x}_{ij}^l)\right]$ and $\mathbb{E}_{\mathcal{D}_u}\left[\mathbf{q}(\mathbf{x}_{ij}^u)\right]$, respectively. Both of these are estimated in the form of EMA in each batch as the training progresses, denoted as $\hat{\mathbb{E}}_{\mathcal{D}_l}\left[\mathbf{q}(\mathbf{x}_{ij}^l)\right]$ and $\hat{\mathbb{E}}_{\mathcal{D}_u}\left[\mathbf{q}(\mathbf{x}_{ij}^u)\right]$. Then we use the ratio between the expectations of labeled and unlabeled to normalize the each prediction $\mathbf{y}_{ij}^u = q(\mathbf{x}_{ij}^u)$ on unlabeled data, formulated as,

$$\text{DA}(\mathbf{y}_{ij}^u) = \text{Norm} \left( \mathbf{y}_{ij}^u \cdot \frac{\hat{\mathbb{E}}_{\mathcal{D}_l}\left[\mathbf{q}(\mathbf{x}_{ij}^l)\right]}{\hat{\mathbb{E}}_{\mathcal{D}_u}\left[\mathbf{q}(\mathbf{x}_{ij}^u)\right]} \right), \quad (12)$$

where $\text{Norm}(\cdot)$ denotes the normalization operation used to constrain the probabilities to sum up to 1. Then we bring the normalized probability back to Equation 2 to calculate the loss weight of each pseudo-label after alignment,

$$\mathcal{L}_u = \frac{1}{N_u} \sum_{i=1}^{N_u} \frac{1}{WH} \sum_{j=1}^{WH} f^*(\max(\text{DA}(\mathbf{y}_{ij}^u))) \cdot \ell_{ce} \left( \hat{\mathbf{y}}_{ij}^u, \mathbf{q}(\mathcal{A}^s(\mathcal{A}^w(\mathbf{x}_{ij}^u))) \right), \quad (13)$$

where $\hat{\mathbf{y}}_{ij}^u = \text{argmax}(\text{DA}(\mathbf{y}_{ij}^u))$. In this way, the distribution-aware weighting function is rewarded with better generalization, benefiting from more equal learning of labeled and unlabeled data, mitigating the issue of distribution imbalance, and enjoying the synergy. The algorithm flow is shown in the supplementary material.

# 3 Experiments

## 3.1 Experimental Setup

**Datasets:** (1) PASCAL VOC 2012 [29] is an object-centric semantic segmentation dataset, containing 21 classes with 1,464 and 1,449 finely annotated images for training and validation, respectively. Some researches [30, 19] augment the original training set (e.g., *classic*) by incorporating the coarsely annotated images in SBD [31], obtaining a training set (e.g., *blender*) with 10,582 labeled samples. (2) Cityscapes [32]is an urban scene understanding dataset with 2,975 images for training and 500 images for validation.

**Implementation Details:** For a fair comparison, we follow the common practice and use ResNet [33] as our backbone and DeepLabv3+[34] as the decoder. We set the crop size as $513 \times 513$ for PASCAL and $801 \times 801$ for Cityscapes, respectively. For both datasets, we adopt SGD as the optimizer with the same batch size of 16 and different initial learing rate, which is set as 0.001 and 0.005 for PASCAL and Cityscapes. We use the polynomial policy to dynamically decay the learning rate along the whole

Table 2: Quantitative results of different SSL methods on Pascal *classic* and *blender* set. We report mIoU (%) under various partition protocols and show the improvements over *Sup.-only* baseline. The **best** is highlighted in **bold**.

| | Method | Classic | | | | | Blender | | |
| --- | --- | --- | --- | --- | --- | --- | --- | --- | --- |
| | | 1/16(92) | 1/8(183) | 1/4(366) | 1/2(732) | Full(1464) | 1/16(662) | 1/8(1323) | 1/4(2646) |
| ResNet-50 | *Sup.-only* | 44.0 | 52.3 | 61.7 | 66.7 | 72.9 | 62.4 | 68.2 | 72.3 |
| | Pseudo-Label[ICML'13] [10] | 55.7 | 60.2 | 65.6 | 69.7 | 74.8 | 66.3 | 70.8 | 74.5 |
| | FixMatch[NeurIPS'20] [18] | 60.1 | 67.3 | 71.4 | 73.7 | 76.9 | 70.6 | 73.9 | 75.1 |
| | iMAS[CVPR'23] [24] | – | – | – | – | – | 74.8 | 76.5 | 77.0 |
| | AugSeg[CVPR'23] [25] | 64.2 | 72.2 | 76.2 | 77.4 | 78.8 | 74.7 | 76.0 | 77.2 |
| | **DAW (Ours)** | **68.5** | **73.1** | **76.3** | **78.6** | **79.7** | **76.2** | **77.6** | **77.4** |
| | Δ ↑ | +24.5 | +20.8 | +14.6 | +11.9 | +6.8 | +13.8 | +9.4 | +5.1 |
| ResNet-101 | *Sup.-only* | 45.1 | 55.3 | 64.8 | 69.7 | 73.5 | 67.5 | 71.1 | 74.2 |
| | Pseudo-Label[ICML'13] [10] | 57.3 | 64.1 | 69.4 | 73.3 | 77.2 | 69.1 | 73.8 | 76.7 |
| | FixMatch[NeurIPS'20] [18] | 63.9 | 73.0 | 75.5 | 77.8 | 79.2 | 74.3 | 76.3 | 76.9 |
| | CPS[CVPR'21] [26] | 64.1 | 67.4 | 71.7 | 75.9 | – | 74.5 | 76.4 | 77.7 |
| | AEL[NeurIPS'21] [27] | – | – | – | – | – | 77.2 | 77.6 | 78.1 |
| | iMAS[CVPR'23] [24] | 68.8 | 74.4 | 78.5 | 79.5 | 81.2 | 76.5 | 77.9 | 78.1 |
| | AugSeg[CVPR'23] [25] | 71.1 | 75.5 | 78.8 | 80.3 | 81.4 | 77.0 | 77.3 | 78.8 |
| | CCVC[CVPR'23] [28] | 70.2 | 74.4 | 77.4 | 79.1 | 80.5 | 77.2 | 78.4 | 79.0 |
| | **DAW (Ours)** | **74.8** | **77.4** | **79.5** | **80.6** | **81.5** | **78.5** | **78.9** | **79.6** |
| | Δ ↑ | +29.7 | +22.1 | +14.7 | +10.9 | +8.0 | +11.0 | +7.8 | +5.4 |

Table 3: Quantitative results of different SSL methods on Cityscapes. We report mIoU (%) under various partition protocols and show the improvements over *Sup.-only* baseline. The **best** is highlighted in **bold**.

| Method | ResNet-50 | | | | ResNet-101 | | | |
| --- | --- | --- | --- | --- | --- | --- | --- | --- |
| | 1/16(186) | 1/8(372) | 1/4(744) | 1/2(1488) | 1/16(186) | 1/8(372) | 1/4(744) | 1/2(1488) |
| *Sup.-only* | 63.3 | 70.2 | 73.1 | 76.6 | 66.3 | 72.8 | 75.0 | 78.0 |
| Pseudo-Label[ICML'13] [10] | 67.2 | 72.4 | 74.9 | 77.4 | 68.9 | 74.3 | 76.8 | 78.6 |
| FixMatch[NeurIPS'20] [18] | 72.6 | 75.7 | 76.8 | 78.2 | 74.2 | 76.2 | 77.2 | 78.4 |
| AEL[NeurIPS'21] [27] | 74.0 | 75.8 | 76.2 | – | 75.8 | 77.9 | 79.0 | 80.3 |
| PCR[NeurIPS'22] [2] | – | – | – | – | 73.4 | 76.3 | 78.4 | 79.1 |
| GTA-Seg[NeurIPS'22] [3] | 63.0 | 69.4 | 72.0 | 76.1 | 69.4 | 72.0 | 76.1 | – |
| iMAS[CVPR'23] [24] | 74.3 | 77.4 | 78.1 | 79.3 | – | – | – | – |
| AugSeg[CVPR'23] [25] | 73.7 | 76.5 | 78.8 | 79.3 | 75.2 | 77.8 | 79.6 | 80.4 |
| **DAW (Ours)** | **75.2** | **77.5** | **79.1** | **79.5** | **76.6** | **78.4** | **79.8** | **80.6** |
| Δ ↑ | +11.9 | +7.3 | +6.0 | +2.9 | +10.3 | +5.6 | +4.8 | +2.6 |

training and assemble the channel dropout perturbation [22] to improve the generalization ability of the model. We train the model for 80 epochs on PASCAL and 240 epochs on Cityscapes, using $8\times$ NVIDIA GeForce RTX 3090 GPUs.

## 3.2 Comparison with State-of-the-art Methods

We conduct experiments on two popular benchmarks including PASCAL VOC 2012 and Cityscapes and make a fair comparison with SOTA semi-supervised semantic segmentation methods. We consistently observe that our DAW outperforms all other methods under all partition protocols on all datasets with different backbones, which strongly proves the effectiveness of our method.

**Results on PASCAL VOC 2012 Dataset.** Table 2 shows the comparison of our method with the SOTA methods on PASCAL *classic* and *blender* set. Specifically, on the PASCAL *classic* set, our method outperforms the supervised-only (*Sup.-only*) model by 29.7%, 22.1%, 14.7%, 10.9% under the partition protocols of 1/16, 1/8, 1/4 and 1/2, respectively with ResNet-101. Our method also significantly outperforms the existing semi-supervised SOTA methods under all data partition protocols. Taking the recently proposed method AugSeg [25] as an example, the performance gain of our approach reaches to +4.3% under 1/16 partition protocol with ResNet-50. The same superiority of our method can also be observed on the PASCAL *blender* set.

**Results on Cityscapes Dataset.** Table 3 compares DAW with SOTA methods on the Cityscapes dataset. DAW achieves consistent performance gains over the *Sup.-only* baseline, obtaining improvements of 11.9%, 7.3%, 6.0% and 2.9% under 1/16, 1/8, 1/4 and 1/2 partition protocols with ResNet-50, respectively. We can also see that over all protocols, DAW outperforms the SOTA methods, e.g., DAW excels to iMAS [24] by 1.1% under the 1/16 partition with ResNet-101.

Table 4: Ablation studies of different components. Note that "Fixed" denotes the result of UniMatch.

| None | Fixed | DAWF | DA | mIoU |
|------|-------|------|-----|------|
| ✓ | | | | 62.7 |
| | ✓ | | | 66.9 |
| | | ✓ | | 68.0 |
| | | ✓ | ✓ | 68.5 |

Table 5: Ablation studies of different momentum on PASCAL *classic* 92.

| $m$ | mIoU |
|-----|------|
| 0.99 | 68.5 |
| 0.999 | 68.1 |
| 0.9999 | 67.9 |

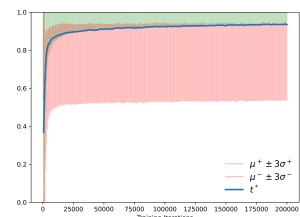

Figure 2: The curve of Pos.&Neg. distribution and $t^*$ during training.

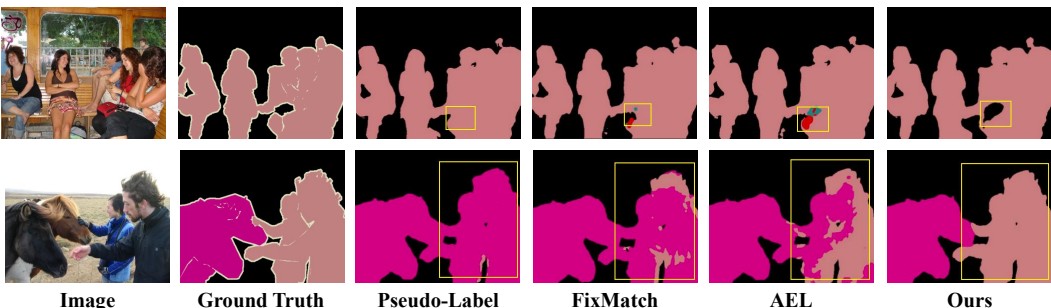

| Image | Ground Truth | Pseudo-Label | FixMatch | AEL | Ours |

Figure 3: Qualitative comparison with different methods. Note that significant improvements are marked with yellow boxes.

**Qualitative Results.** We compare qualitative results of our DAW with different SOTA methods. As shown in Figure 3 , DAW also shows more powerful segmentation performance in fine-grained details (see the first and second row in Figure 3). With the help of the optimal weighting function, DAW demonstrates superior abilities in most scenarios.

### 3.3 Ablation Study and Analysis

To look deeper into our method, we perform a series of ablation studies on PASCAL *classic* set under 92 partition protocol with ResNet-50 to analyze each component of our DAW, including the **D**istribution-**A**ware **W**eighting **F**unction (DAWF) and the **D**istribution **A**lignment (DA). The baseline method is UniMatch [22].

**Effectiveness of Components.** In Table 4, "None" denotes there is no threshold for pseudo-label during the training (i.e., Pseudo-Label [10]) while "Fixed" indicates that a fixed threshold is set (i.e., UiMatch [22]). A certain performance lift compared with the baseline can be observed owing to

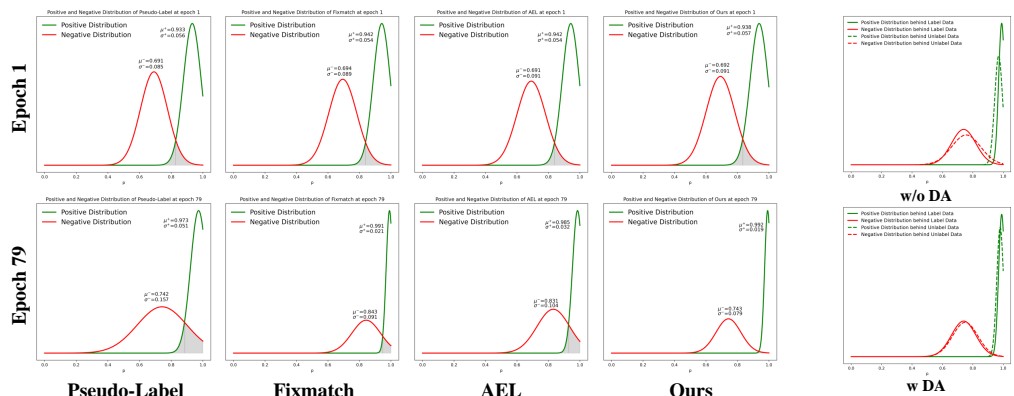

Figure 4: Comparison between distributions of different methods at different training epochs (e.g., epoch 1 *vs.* epoch 79).

Figure 5: Viz. of Dist. w&w/o DA.

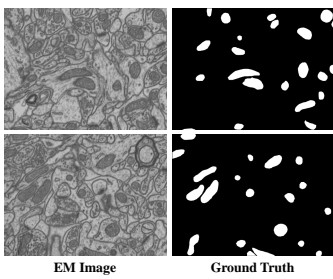

**EM Image**      **Ground Truth**

Figure 6: Visualization of Lucchi dataset.

Table 6: Quantitative results of different SSL methods on Lucchi dataset. We report mIoU (%) under various partition protocols. The **best** is highlighted in bold.

| Method | 1/32(5) | 1/16(10) | 1/8(20) |
|---|---|---|---|
| *Sup.-only* | 45.7 | 57.4 | 61.8 |
| MT [35] | 71.8 | 72.4 | 75.4 |
| CCT [36] | 84.7 | 85.4 | 85.8 |
| CPS [30] | 84.5 | 84.6 | 85.8 |
| **DAW (Ours)** | **85.9** | **86.6** | **87.6** |

the introduction of Distribution-Aware Weighting Function and Distribution Alignment. (1) The utilization of DAWF brings a 1.1% improvement of mIoU, demonstrating that the negative impact raised by the $E1 + E2$ trade-off is effectively alleviated. (2) DA brings further accuracy gains, indicating that the existence of the discrepancy between the distributions of labeled and unlabeled data may cause a bottleneck in learning. For better visualization, we employ the unused annotations of "unlabeled data" to calculate the ground-truth distribution on the unlabeled data. And as shown in Figure 5, there is a relatively large gap between the distributions, and DA can effectively relieve it, further unlocking the potential of the weighting function and enjoy the synergy.

**Hyperparameter Evaluations.** As shown in Table 5, it can be observed that the performance is optimal with $m = 0.99$.

**Scalability for Other Scenarios.** We further conduct extra experiments on Lucchi [37–42] to evaluate the scalability of our method. Figure 6 shows the image and ground-truth of Lucchi dataset, presenting a common problem in electron microscope images that the instances are very small and scattered. This calls for more reliable supervision in training under a semi-supervised setting. As shown in Table 6, DAW outperforms other competitive methods in the electron microscopy domain, indicating that our method can provide more reliable supervision.

**Comparison of Distribution.** As shown in Figure 4, the distributions of different methods are almost the same. As the learning goes on, the discrepancy between the positive and negative distributions of ours becomes larger (simultaneously shown in Figure 2) while the others almost no change. A large discrepancy between the positive and negative distributions means that we can filter out as many negative samples while recruiting as many positive samples as possible, which is conducive to model training. This is the fundamental reason behind why our method outperforms other methods.

## 4 Related Work

**Semi-Supervised Learning.** Semantic segmentation is a fundamental task that has achieved conspicuous achievements credited to the recent advances in deep neural networks [43–48]. However, its data-driven nature makes it heavily dependent on massive pixel-level annotations, which are laborious and time-consuming to gather. To alleviate the data-hunger issue, considerable works have turned their attention to semi-supervised learning. Designing appropriate and effective supervision signals for unlabelled data is the core problem of semi-supervised learning. Previous research can be summarized into two learning schemes: self-training and consistency regularization. Self-training based methods [49, 12, 10, 50] aim to train the model based on the pseudo-labels generated by the up-to-date optimized model for the unlabelled data. Consistency regularization-based methods aim to obtain prediction invariance under various perturbations, including input perturbation [51, 52], feature perturbation [53] and network perturbation [54–57]. The recently semi-supervised methods, MixMathch [58] and FixMatch [18] combine these two techniques together and achieve state-of-the-art performance. Based on the paradigm of existing semi-supervised learning methods, our method explores a better weighting function for the pseudo-label scheme during training.

**Semi-Supervised Semantic Segmentation.** Semi-supervised semantic segmentation aims at pixel-level classification with limited labeled data. Recently, following the paradigm of semi-supervised learning, many semi-supervised semantic segmentation methods also focus on the design of self-training [26, 27, 5] and consistency regularization [59, 60, 53, 61] strategies. U$^2$PL [5] proposes to

make sufficient use of unreliable pseudo-labeled data. CCT [53] adopts a feature-level perturbation and enforces consistency among the predictions from different decoders. More recently, SOTA semi-supervised segmentation methods also integrate both technologies for better performance. PseudoSeg [7], AEL [27] and UCC [62] propose to use the pseudo-labels generated from weak augmented images to constrain the predictions of strong augmented images. In this paper, we shed light on semi-supervised semantic segmentation based on pseudo-labeling and strive to explore better strategies for using pseudo-labels.

## 5   Conclusion

In this paper, we propose DAW to systematically analyze the trade-off in previous methods that hinder the model's learning. We formally define the trade-off between inaccurate yet utilized pseudo-labels, and correct yet discarded pseudo-labels by explicitly modeling the confidence distribution of correct and inaccurate pseudo-labels, equipped with a unified weighting function. Experiments show the effectiveness.

## 6   Acknowledgments

This work was partially supported by the National Defense Basic Scientific Research Program (Grant JCKY2020903B002).

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
