# DAW: Exploring the Better Weighting Function for Semi-supervised Semantic Segmentation

## *Supplementary Material*

**Rui Sun**[1*]     **Huayu Mai**[1*]     **Tianzhu Zhang**[1,2†]     **Feng Wu**[1,2]

[1]Deep Space Exploration Laboratory/School of Information Science and Technology,
University of Science and Technology of China

[2]Institute of Artificial Intelligence, Hefei Comprehensive National Science Center

`{issunrui, mai556}@mail.ustc.edu.cn`, `{tzzhang, fengwu}@ustc.edu.cn`

In the supplementary material, we first introduce the pseudo algorithm of DAW. Then we clarify the $E1$-$E2$ trade-off for unlabeled data and perform more analysis for Figures 1, 3, 4 and 5. Furthermore, we analyze the reason for choosing the Gaussian distribution. Next, we present the detailed derivations in Table 1. Then, we formulate the optimal weighting function by solving the optimization problem through functional analysis. Finally, we show more qualitative results for DAW compared to other competitors.

## 6   Algorithm

In this section, we present the pseudo algorithm of DAW to clearly summarize the method of the main paper (Section 2). DAW formalizes the selection criteria for reliable pseudo-labels from unlabeled data in a dynamic manner at each training iteration, encapsulating distribution-aware weighting function (DAWF) for minimizing the negative equivalence impact raised by the trade-off ($E1$-$E2$), and distribution alignment (DA) for alleviating the gap in confidence distribution caused by the discrepancy in prediction distribution between labeled and unlabeled data. Please refer to Section 2 of the main paper for detailed elaboration.

## 7   E1-E2 Trade-off for Unlabeled Data and More for Figures 1, 2, 4 and 5

In this section, we describe in detail how to calculate the $E1$ and $E2$ which reflect the trade-off for unlabeled data, considering that the calculated weighting function $f^*(p)$ is based on positive and negative confidence distributions $g^+(p)/g^-(p)$, which are determined by the labeled data and their corresponding ground truth. Then, we provide a more detailed explanation of Figures 1, 2, 4, and 5, which are slightly abbreviated due to the limited space of the main paper.

### 7.1   E1-E2 Trade-off for Unlabeled Data and More for Figure 5

To verify the effectiveness of our distribution-aware weighting function (DAWF) on unlabeled data, we formalize the trade-off for unlabeled data, including inaccurate yet utilized pseudo-labels ($E1$), and correct yet discarded pseudo-labels ($E2$). In practice, $E1$ is the proportion of inaccurate yet utilized pseudo-labels in all inaccurate pseudo-labels; $E2$ equals the proportion of correct yet discarded pseudo-labels in all correct pseudo-labels, assuming that the ground truth for unlabeled data is available, which can determine the correctness of the pseudo-labels, solely for theoretical analysis purposes.

---

*Equal contribution

†Corresponding author

Preprint. Under review.

---

**Algorithm 1** Pseudo algorithms of DAW.

---

1: **Input:** Labeled set $\mathcal{D}_l = \left\{ \mathbf{x}_i^l, \mathbf{y}_i^l \right\}_{i=1}^{N_L}$, Unlabeled set $\mathcal{D}_u = \left\{ \mathbf{x}_i^u \right\}_{i=1}^{N_U}$ ($N_U \gg N_L$).

2: **Define:** Prediction $\mathbf{q}(\mathbf{x}_{ij})$, $\hat{\mathbf{y}}_{ij}^l = \operatorname{argmax}(\mathbf{q}(\mathbf{x}_{ij}^l))$, $\mathbf{y}_{ij}^u = \mathbf{q}(\mathcal{A}^w(\mathbf{x}_{ij}^u))$

3: **for** each batch (training iteration $t$) of $\left\{ \mathbf{x}_i^l, \mathbf{y}_i^l \right\}_{i=1}^{N_l}$, $\left\{ \mathbf{x}_i^u \right\}_{i=1}^{N_u}$ in $\mathcal{D}_l, \mathcal{D}_u$ ($N_l = N_u$) **do**

4:   *# Labeled Data:*

5:   Calculate $\mathcal{L}_s$ by Equation 1              ▷ *Supervised loss*

6:   Select correct prediction from labeled data: $\hat{\mathbf{y}}_{ij}^l = \mathbf{y}_{ij}^l$

7:   Select inaccurate prediction from labeled data: $\hat{\mathbf{y}}_{ij}^l \neq \mathbf{y}_{ij}^l$

8:   Calculate $\hat{\mu^+}, \hat{\sigma^+}$ and $\hat{\mu^-}, \hat{\sigma^-}$ from correct and inaccurate prediction by Equation 5

9:   EMA update the $\hat{\mu_t^+}, \hat{\sigma_t^+}$ and $\hat{\mu_t^-}, \hat{\sigma_t^-}$ to fit $g^+(p)$ and $g^-(p)$ by Equation 6

10:   Determine the $f^*(p)$ based on the updated $g^+(p)$ and $g^-(p)$ by Equation 11    ▷ *DAWF*

11:   *# Unlabeled Data:*

12:   Modulate $\mathbf{y}_{ij}^u$ by Equation 12                 ▷ *DA*

13:   Calculate $\mathcal{L}_u$ based on $f^*(p)$ by Equation 13       ▷ *Unsupervised loss*

14:   Gradient backward $\mathcal{L}_s + \mathcal{L}_u$           ▷ *Update Model*

15: **end for**

---

Note that there may exist a gap between the prediction distribution of labeled and unlabeled data, which hinders the generalization of the generated weighting function $f^*(p)$. Therefore, we propose distribution alignment (DA) to alleviate this gap, aiming to further unlock the potential of the weighting function, that is, realizing the effective discrimination of the reliable pseudo-labels (see Section 2.4 and Figure 5).

### 7.2 More Analysis of Figures 1, 2 and 4

Based on the definitions of $E1$ and $E2$ on unlabeled data, we can easily compare DAW with other methods. Specifically, Figure 4 showcases the variations in the positive and negative confidence distributions of different methods as the training progresses, where the overlapping region of the distributions (indicated as the gray solid region) reflects the separation of the two distributions. We can clearly observe that at the beginning of the training, the distributions of different methods are almost identical. However, as the training progresses, benefiting from the proposed distribution-aware weighting function that minimizes the negative equivalence impact of the trade-off, the model improves. And in turn, DAW exhibits the maximum separation between the positive and negative confidence distributions compared to other methods (Figure 4), leading to a simultaneous decrease in both $E1$ and $E2$, which is consistent with the observation of Figure 1.

To vividly present the working mechanism of the distribution-aware weighting function (DAWF), we visualize the variations in positive and negative confidence distribution ($3\sigma$ statistical rule) as well as the jump point of the $f^*(p)$ as the training progresses, as illustrated in Figure 2. We can observe that $f^*(p)$ strives to filter out as many negative samples while recruiting as many positive samples as possible, which is conducive to model training. This is the fundamental reason behind that our method outperforms other methods (Please refer to the "*Comparison of Distribution paragraph*" of the main paper in Section 3.3).

## 8 Reason for Choosing Gaussian

In this section, we explain why we choose Gaussian distribution from the perspective of the principle of maximum entropy. Since there is no prior about the positive/negative distribution, we should choose the distribution with the maximum entropy to minimize risk, which is stated by the principle of maximum entropy. Given the mean $\mu$ and variance $\sigma^2$ estimated from the observed samples of

distribution $p(x)$, searching the distribution with maximum entropy are equivalent to the following optimization problem:

$$\max_{p(x)} H(p) = -\int_{-\infty}^{\infty} p(x) \log p(x)\, dx,$$

$$\text{s.t.} \begin{cases} p(x) \geq 0 \\[2mm] \int_{-\infty}^{\infty} p(x)\, dx = 1 \\[2mm] \int_{-\infty}^{\infty} x p(x)\, dx = \mu \\[2mm] \int_{-\infty}^{\infty} (x-\mu)^2 p(x)\, dx = \sigma^2 \end{cases} \tag{14}$$

To solve the above optimization problem, the following Lagrangian equation is constructed:

$$J(p) = -\int p(x) \log p(x)\, dx + \lambda_0 \int p(x)\, dx + \lambda_1 \int x p(x)\, dx + \lambda_2 \int (x-\mu)^2 p(x)\, dx. \tag{15}$$

The constrained optimization problem 14 is converted into $\max_p J(p)$, which can be solved by calculating variations.

We take the derivative of $J(p)$ with respect to $p(x)$ and the derivative should be 0:

$$\frac{\partial J(p)}{\partial p} = -\log p(x) - 1 + \lambda_0 + \lambda_1 x + \lambda_2 (x-\mu)^2 = 0. \tag{16}$$

Therefore, we can get $p(x)$ as:

$$p(x) = e^{-1+\lambda_0+\lambda_1 x+\lambda_2 (x-\mu)^2}. \tag{17}$$

Substituting Equation 17 back into the constraints, we get:

$$\begin{cases} \int_{-\infty}^{\infty} e^{-1+\lambda_0+\lambda_1 x+\lambda_2 (x-\mu)^2}\, dx = 1 \\[2mm] \int_{-\infty}^{\infty} x e^{-1+\lambda_0+\lambda_1 x+\lambda_2 (x-\mu)^2}\, dx = \mu \\[2mm] \int_{-\infty}^{\infty} (x-\mu)^2 e^{-1+\lambda_0+\lambda_1 x+\lambda_2 (x-\mu)^2}\, dx = \sigma^2 \end{cases} \implies p(x) = \frac{1}{\sqrt{2\pi}\sigma} e^{-\frac{(x-\mu)^2}{2\sigma^2}}. \tag{18}$$

It can be seen that the distribution of the maximum entropy that satisfies the constraints is a Gaussian distribution, whose mean is $\mu$ and variance is $\sigma$.

## 9 Derivations of Table 1

In this section, we present the detailed derivations for $E1, E2, E1 + E2$ of different methods in Table 1. In the main paper, we formally define the trade-off between inaccurate yet utilized pseudo-labels ($E1$), and correct yet discarded pseudo-labels ($E2$) by explicitly modeling the confidence distribution of correct ($g^+(p)$, positive distribution) and inaccurate ($g^-(p)$, negative distribution) pseudo-labels, equipped with a unified weighting function ($f(p)$), formulated as follows:

$$g^+(p) = \begin{cases} \frac{\alpha}{\sqrt{2\pi}\sigma^+} \exp\left[-\frac{(p-\mu^+)^2}{2(\sigma^+)^2}\right], & \frac{1}{C} \leqslant p \leqslant 1 \\[3mm] 0, & \text{otherwise} \end{cases}, \tag{19}$$

$$g^-(p) = \begin{cases} \frac{\beta}{\sqrt{2\pi}\sigma^-} \exp\left[-\frac{(p-\mu^-)^2}{2(\sigma^-)^2}\right], & \frac{1}{C} \leqslant p \leqslant 1 \\[3mm] 0, & \text{otherwise} \end{cases}, \tag{20}$$

$$E1 = \mathbb{E}_{g^-}[f(p)] = \int_{\frac{1}{C}}^{1} f(p) \cdot g^-(p)\, dp, \tag{21}$$

$$E2 = 1 - \mathbb{E}_{g^+}[f(p)] = 1 - \int_{\frac{1}{C}}^{1} f(p) \cdot g^+(p)\, dp, \tag{22}$$

$$E1 + E2 = 1 + \int_{\frac{1}{C}}^{1} f(p) \cdot [g^-(p) - g^+(p)]\, dp. \tag{23}$$

In practice, there must exist one intersection $t^*$ of $g^+(p)$ and $g^-(p)$ between $\left[\frac{1}{C}, 1\right]$, where satisfies:

$$\begin{cases} g^-(p) > g^+(p), & \frac{1}{C} < p < t^* \\ g^-(p) < g^+(p), & t^* < p < 1 \end{cases}. \tag{24}$$

Thus we can decompose Equation 23 into two integrals:

$$E1 + E2 = 1 + \int_{\frac{1}{C}}^{t^*} f(p) \cdot \left[g^-(p) - g^+(p)\right] dp + \int_{t^*}^{1} f(p) \cdot \left[g^-(p) - g^+(p)\right] dp. \tag{25}$$

In the following, we instantiate the above formulas for each relevant method.

## 9.1 Pseudo-Label

In the naive pseudo-labeling method, all pseudo labels are enrolled into training, i.e., $f(p) = u(p)$. Thus we have:

$$\begin{aligned} E1 &= \int_{\frac{1}{C}}^{1} u(p) \cdot g^-(p)\, dp \\ &= \int_{\frac{1}{C}}^{1} g^-(p)\, dp \\ &= 1 \end{aligned} \tag{26}$$

$$\begin{aligned} E2 &= 1 - \int_{\frac{1}{C}}^{1} u(p) \cdot g^+(p)\, dp \\ &= 1 - \int_{\frac{1}{C}}^{1} g^+(p)\, dp \\ &= 1 - 1 \\ &= 0 \end{aligned} \tag{27}$$

which demonstrates that maximizing the utilization of correct pseudo-labels ($E2 = 0$) inevitably absorbs a lot of inaccurate pseudo-labels ($E1 = 1$) in training, leading to confirmation bias.

$E1 + E2$

$$\begin{aligned} &= 1 + \int_{\frac{1}{C}}^{t^*} u(p) \cdot \left[g^-(p) - g^+(p)\right] dp + \int_{t^*}^{1} u(p) \cdot \left[g^-(p) - g^+(p)\right] dp \\ &= 1 + \int_{\frac{1}{C}}^{t^*} \left[g^-(p) - g^+(p)\right] dp + \int_{t^*}^{1} \left[g^-(p) - g^+(p)\right] dp \\ &= 1 + \beta \left[\Phi(\frac{1 - \mu^-}{\sigma^-}) - \Phi(\frac{t^* - \mu^-}{\sigma^-})\right] - \alpha \left[\Phi(\frac{1 - \mu^+}{\sigma^+}) - \Phi(\frac{t^* - \mu^+}{\sigma^+})\right] + \int_{\frac{1}{C}}^{t^*} \left[g^-(p) - g^+(p)\right] dp \\ &= 1 \end{aligned} \tag{28}$$

Since $g^-(p) > g^+(p)$ when $\frac{1}{C} < p < t^*$, we have:

$$\int_{\frac{1}{C}}^{t^*} \left[ g^-(p) - g^+(p) \right] dp > 0. \tag{29}$$

## 9.2 FixMatch

For threshold-based pseudo-labeling methods such as FixMatch, which attempts to set a high threshold ($t = 0.95$), i.e., $f(p) = u(p - t)$. Thus we have:

$$
\begin{aligned}
E1 &= \int_{\frac{1}{C}}^{1} u(p - t) \cdot g^-(p) \, dp \\
&= \int_{t}^{1} g^-(p) \, dp \\
&= \beta \left[ \Phi(\frac{1 - \mu^-}{\sigma^-}) - \Phi(\frac{t - \mu^-}{\sigma^-}) \right]
\end{aligned}
\tag{30}
$$

$$
\begin{aligned}
E2 &= 1 - \int_{\frac{1}{C}}^{1} u(p - t) \cdot g^+(p) \, dp \\
&= 1 - \int_{t}^{1} g^+(p) \, dp \\
&= 1 - \alpha \left[ \Phi(\frac{1 - \mu^+}{\sigma^+}) - \Phi(\frac{t - \mu^+}{\sigma^+}) \right]
\end{aligned}
\tag{31}
$$

It can be seen that although setting a high threshold can filter out most inaccurate pixel-level pseudo-labels with low confidence (small value of $E1$ caused by the proximity of $t = 0.95$ to 1), the strict criteria inevitably lead to the contempt of numerous unconfident yet correct pseudo-labels (large value of $E2$ caused by the proximity of $t = 0.95$ to 1).

$E1 + E2$

$$
\begin{aligned}
&= 1 + \int_{\frac{1}{C}}^{t^*} u(p - t) \cdot \left[ g^-(p) - g^+(p) \right] dp + \int_{t^*}^{1} u(p - t) \cdot \left[ g^-(p) - g^+(p) \right] dp \\
&= 1 + \int_{t}^{1} \left[ g^-(p) - g^+(p) \right] dp \\
&= 1 + \int_{t^*}^{1} \left[ g^-(p) - g^+(p) \right] dp - \int_{t^*}^{t} \left[ g^-(p) - g^+(p) \right] dp \\
&= 1 + \beta \left[ \Phi(\frac{1 - \mu^-}{\sigma^-}) - \Phi(\frac{t^* - \mu^-}{\sigma^-}) \right] - \alpha \left[ \Phi(\frac{1 - \mu^+}{\sigma^+}) - \Phi(\frac{t^* - \mu^+}{\sigma^+}) \right] - \int_{t^*}^{t} \left[ g^-(p) - g^+(p) \right] dp \\
&= 1 + \beta \left[ \Phi(\frac{1 - \mu^-}{\sigma^-}) - \Phi(\frac{t - \mu^-}{\sigma^-}) \right] - \alpha \left[ \Phi(\frac{1 - \mu^+}{\sigma^+}) - \Phi(\frac{t - \mu^+}{\sigma^+}) \right]
\end{aligned}
\tag{32}
$$

Since $g^-(p) < g^+(p)$ when $t^* < p < t$, we have:

$$- \int_{t^*}^{t} \left[ g^-(p) - g^+(p) \right] dp \geqslant 0, \tag{33}$$

where equality is obtained at $t = t^*$.

## 9.3 AEL

As a compromise, AEL ad hoc defines the weighting function as a power function, which assigns weights conditioned on confidence, i.e., $f(p) = p^2$. Thus we have:

$$E1 = \int_{\frac{1}{C}}^{1} p^2 \cdot g^-(p)\, dp$$

$$= \int_{\frac{1}{C}}^{1} (p - \mu^-)^2 \cdot g^-(p)\, dp + 2\mu^- \int_{\frac{1}{C}}^{1} p \cdot g^-(p)\, dp - (\mu^-)^2 \int_{\frac{1}{C}}^{1} g^-(p)\, dp$$

$$= -(\sigma^-)^2 \int_{\frac{1}{C}}^{1} (p - \mu^-)\, dg^-(p) + 2\mu^- \left[ \int_{\frac{1}{C}}^{1} (p - \mu^-) \cdot g^-(p)\, dp + \mu^- \int_{\frac{1}{C}}^{1} g^-(p)\, dp \right] - (\mu^-)^2$$

$$= -(\sigma^-)^2 \left[ (p - \mu^-)g^-(p)\Big|_{\frac{1}{C}}^{1} - \int_{\frac{1}{C}}^{1} g^-(p)\, dp \right] + 2\mu^- \left[ -(\sigma^-)^2 \int_{\frac{1}{C}}^{1} dg^-(p) + \mu^- \right] - (\mu^-)^2$$

$$= -(\sigma^-)^2 \left[ (1 - \mu^-)g^-(1) - (\frac{1}{C} - \mu^-)g^-(\frac{1}{C}) - 1 \right] + 2\mu^- \left[ -(\sigma^-)^2 g^-(p)\Big|_{\frac{1}{C}}^{1} + \mu^- \right] - (\mu^-)^2$$

$$= (\mu^-)^2 + (\sigma^-)^2 + \mu^-(\sigma^-)^2 g^-(\frac{1}{C}) - (1 + \mu^-)(\sigma^-)^2 g^-(1)$$

$$\tag{34}$$

$$E2 = 1 - \int_{\frac{1}{C}}^{1} p^2 \cdot g^+(p)\, dp$$

$$= 1 - \int_{\frac{1}{C}}^{1} (p - \mu^+)^2 \cdot g^+(p)\, dp - 2\mu^+ \int_{\frac{1}{C}}^{1} p \cdot g^+(p)\, dp + (\mu^+)^2 \int_{\frac{1}{C}}^{1} g^+(p)\, dp$$

$$= 1 - (\sigma^+)^2 \int_{\frac{1}{C}}^{1} (p - \mu^+)\, dg^+(p) - 2\mu^+ \left[ \int_{\frac{1}{C}}^{1} (p - \mu^+) \cdot g^+(p)\, dp + \mu^+ \int_{\frac{1}{C}}^{1} g^+(p)\, dp \right] + (\mu^+)^2$$

$$= 1 - (\sigma^+)^2 \left[ (p - \mu^+)g^+(p)\Big|_{\frac{1}{C}}^{1} - \int_{\frac{1}{C}}^{1} g^-(p)\, dp \right] - 2\mu^+ \left[ -(\sigma^+)^2 \int_{\frac{1}{C}}^{1} dg^+(p) + \mu^+ \right] + (\mu^+)^2$$

$$= 1 - (\sigma^+)^2 \left[ (1 - \mu^+)g^+(1) - (\frac{1}{C} - \mu^+)g^+(\frac{1}{C}) - 1 \right] - 2\mu^+ \left[ -(\sigma^+)^2 g^+(p)\Big|_{\frac{1}{C}}^{1} + \mu^+ \right] + (\mu^+)^2$$

$$= 1 - (\mu^+)^2 - (\sigma^+)^2 - \mu^+(\sigma^+)^2 g^+(\frac{1}{C}) + (1 - \mu^+)(\sigma^+)^2 g^+(1)$$

$$\tag{35}$$

$E1 + E2$

$$= 1 + \int_{\frac{1}{C}}^{t^*} p^2 \cdot \left[ g^-(p) - g^+(p) \right]\, dp + \int_{t^*}^{1} p^2 \cdot \left[ g^-(p) - g^+(p) \right]\, dp$$

$$= 1 + \int_{t^*}^{1} \left[ g^-(p) - g^+(p) \right]\, dp + \int_{\frac{1}{C}}^{t^*} p^2 \cdot \left[ g^-(p) - g^+(p) \right]\, dp + \int_{t^*}^{1} (p^2 - 1) \cdot \left[ g^-(p) - g^+(p) \right]\, dp$$

$$= 1 + \beta \left[ \Phi(\frac{1 - \mu^-}{\sigma^-}) - \Phi(\frac{t^* - \mu^-}{\sigma^-}) \right] - \alpha \left[ \Phi(\frac{1 - \mu^+}{\sigma^+}) - \Phi(\frac{t^* - \mu^+}{\sigma^+}) \right]$$

$$+ \int_{\frac{1}{C}}^{t^*} p^2 \cdot \left[ g^-(p) - g^+(p) \right]\, dp + \int_{t^*}^{1} (p^2 - 1) \cdot \left[ g^-(p) - g^+(p) \right]\, dp$$

$$= 1 - (\mu^+)^2 - (\sigma^+)^2 - \mu^+(\sigma^+)^2 g^+(\frac{1}{C}) + (1 - \mu^+)(\sigma^+)^2 g^+(1)$$

$$+ (\mu^-)^2 + (\sigma^-)^2 + \mu^-(\sigma^-)^2 g^-(\frac{1}{C}) - (1 + \mu^-)(\sigma^-)^2 g^-(1)$$

$$\tag{36}$$

Since $g^-(p) > g^+(p)$ when $\frac{1}{C} < p < t^*$ and $g^-(p) < g^+(p), p^2 < 1$ when $t^* < p < 1$, we have:

$$\int_{\frac{1}{C}}^{t^*} p^2 \cdot \left[g^-(p) - g^+(p)\right] \, dp + \int_{t^*}^1 (p^2 - 1) \cdot \left[g^-(p) - g^+(p)\right] \, dp > 0. \tag{37}$$

The lack of sophisticated consideration and the arbitrary power for the weighting function inevitably compromise its capability (not guaranteeing the lowest negative equivalence effect).

### 9.4 Ours

By leveraging functional analysis, we find that the optimal weighting function is a step function with the jump point located at the intersection of positive and negative distributions, i.e., $f^*(p) = u(p - t^*)$. Thus we have:

$$\begin{aligned}
E1 &= \int_{\frac{1}{C}}^1 u(p - t^*) \cdot g^-(p) \, dp \\
&= \int_{t^*}^1 g^-(p) \, dp \\
&= \beta \left[\Phi(\frac{1 - \mu^-}{\sigma^-}) - \Phi(\frac{t^* - \mu^-}{\sigma^-})\right]
\end{aligned} \tag{38}$$

$$\begin{aligned}
E2 &= 1 - \int_{\frac{1}{C}}^1 u(p - t^*) \cdot g^+(p) \, dp \\
&= 1 - \int_{t^*}^1 g^+(p) \, dp \\
&= 1 - \alpha \left[\Phi(\frac{1 - \mu^+}{\sigma^+}) - \Phi(\frac{t^* - \mu^+}{\sigma^+})\right]
\end{aligned} \tag{39}$$

$$\begin{aligned}
&E1 + E2 \\
&= 1 + \int_{\frac{1}{C}}^{t^*} u(p - t^*) \cdot \left[g^-(p) - g^+(p)\right] \, dp + \int_{t^*}^1 u(p - t^*) \cdot \left[g^-(p) - g^+(p)\right] \, dp \\
&= 1 + \int_{t^*}^1 \left[g^-(p) - g^+(p)\right] \, dp \\
&= 1 + \beta \left[\Phi(\frac{1 - \mu^-}{\sigma^-}) - \Phi(\frac{t^* - \mu^-}{\sigma^-})\right] - \alpha \left[\Phi(\frac{1 - \mu^+}{\sigma^+}) - \Phi(\frac{t^* - \mu^+}{\sigma^+})\right]
\end{aligned} \tag{40}$$

As shown in Equation 29, Equation 33 and Equation 37, our weighting function achieves the minimum $E1 + E2$, which is guaranteed by theoretical functional analysis in the next section.

## 10 Proof of Optimal Weighting Function

In this section, we solve the optimization problem by the variational calculus derived from functional analysis, providing a solid theoretical basis for our approach. To solve the optimization problem:

$$\begin{aligned}
\min_{f(p)} \quad & E1 + E2 = 1 + \int_{\frac{1}{C}}^1 f(p) \cdot \left[g^-(p) - g^+(p)\right] \, dp, \\
\text{s.t.} \quad & 0 \le f(p) \le 1.
\end{aligned} \tag{41}$$

We first define a functional $J[f]$ as:

$$J[f] = \int_{\frac{1}{C}}^1 f(p) \cdot g(p) \, dp, \tag{42}$$

where $g(p) = g^-(p) - g^+(p)$. We use variational calculus to find the optimal $f^*(p)$ minimizing the $J[f]$ subject to the constraints that $f(p)$ is bounded by $[0, 1]$. Considering an arbitrary integrable function $\delta f(p)$, we have:

$$J[f + \varepsilon \delta f] = \int_{\frac{1}{C}}^{1} [f(p) + \varepsilon \delta f(p)] \cdot g(p)\, dp$$

$$= \int_{\frac{1}{C}}^{1} f(p) \cdot g(p)\, dp + \varepsilon \int_{\frac{1}{C}}^{1} \delta f(p) \cdot g(p)\, dp, \tag{43}$$

where $\varepsilon$ is a small quantity. We can derive the $\delta J[f]$ by taking the limit:

$$\delta J[f] = \lim_{\varepsilon \to 0} \frac{J[f + \varepsilon \delta f] - J[f]}{\varepsilon}$$

$$= \int_{\frac{1}{C}}^{1} \delta f(p) \cdot g(p)\, dp. \tag{44}$$

Assuming that we reach the minimum $J[f]$ at $f^*(p)$, this requires $\delta J[f^*]$ to be always greater than zero ($\delta J[f^*] > 0$) for arbitrary $\delta f(p)$, i.e.,

$$\int_{\frac{1}{C}}^{1} \delta f(p) \cdot g(p)\, dp > 0. \tag{45}$$

For a general $f(x)$, due to the relaxed constraints ($0 \leq f(p) \leq 1$), $\delta f(p)$ has almost no restrictions, making it challenging to satisfy the above condition. Indeed, when there is a constraint on $\delta f(p)$ that:

$$\begin{cases} \delta f(p) > 0, & g(p) > 0 \\ \delta f(p) < 0, & g(p) < 0 \end{cases}, \tag{46}$$

Inequality 45 holds true at all times. The analogous constraint for $\delta f(p)$ can be obtained when the $f(p)$ stays at its bound ($0/1$); this is because there is a constraint that $0 \leq f(p) + \delta f(p) \leq 1$. That is to say:

$$f^*(p) = \begin{cases} 0, & g^-(p) - g^+(p) > 0 \\ 1, & g^-(p) - g^+(p) < 0 \end{cases}. \tag{47}$$

In practice, it is observed that $g^-(p)$ and $g^+(p)$ intersect at a point $t^*$ within the range of $\left[\frac{1}{C}, 1\right]$, satisfying equation 24. Thus the optimal weighting function $f^*(p)$ is a step function jumping at $t^*$, i.e., $f^*(p) = u(p - t^*)$.

## 11  Exact Value of $t^*$

In this section, we get the exact value of $t^*$ by solving the quadratic equation. As proofed above, the step function jumping at the intersection of the positive distribution $g^+(p)$ and the negative distribution $g^-(p)$ is the optimal weighting function. In this section, we calculate the exact value of $t^*$ (only considering the range of $[\frac{1}{C}, 1]$):

$$g^+(p) = g^-(p)$$

$$\frac{\alpha}{\sqrt{2\pi}\sigma^+} \exp\left[-\frac{(p - \mu^+)^2}{2(\sigma^+)^2}\right] = \frac{\beta}{\sqrt{2\pi}\sigma^-} \exp\left[-\frac{(p - \mu^-)^2}{2(\sigma^-)^2}\right] \tag{48}$$

$$\ln \frac{\alpha \sigma^-}{\beta \sigma^+} - \frac{(p - \mu^+)^2}{2(\sigma^+)^2} = -\frac{(p - \mu^-)^2}{2(\sigma^-)^2}$$

The same type of term is combined to obtain:

$$\left[(\sigma^+)^2 - (\sigma^-)^2\right] p^2 + 2\left[\mu^+(\sigma^-)^2 - \mu^-(\sigma^+)^2\right] p + \left[(\sigma^+\mu^-)^2 - (\sigma^-\mu^+)^2 + 2(\sigma^+\sigma^-)^2 \ln \frac{\alpha\sigma^-}{\beta\sigma^+}\right]$$

$$= 0 \tag{49}$$

Let $\beta_1$, $\beta_2$ and $\beta_3$ as followed:

$$\beta_1 = (\sigma^+)^2 - (\sigma^-)^2, \tag{50}$$

$$\beta_2 = 2\left[\mu^+(\sigma^-)^2 - \mu^-(\sigma^+)^2\right], \tag{51}$$

$$\beta_3 = (\sigma^+\mu^-)^2 - (\sigma^-\mu^+)^2 + 2(\sigma^+\sigma^-)^2\ln\frac{\alpha\sigma^-}{\beta\sigma^+}. \tag{52}$$

In practice, $\sqrt{\beta_2^2 - 4\beta_1\beta_3} > 0$ and $\sigma^+ < \sigma^-$, so $\beta_1 < 0$. To satisfy the requirement of $t^* \in \left[\frac{1}{C}, 1\right]$, we have:

$$t^* = \frac{-\beta_2 + \sqrt{\beta_2^2 - 4\beta_1\beta_3}}{2\beta_1}, \tag{53}$$

## 12 More Qualitative Result

In this section, we provide more qualitative results between ours and other competitors. As shown in Figure 7, other methods are easily distracted by adjacent targets, resulting in confusing segmentation errors, e.g., in the first row of Figure 7, other existing methods have difficulty distinguishing between *dog* and *cat* that are close together, while our method can identify both in a relatively complete way, which demonstrates the effectiveness of our method.

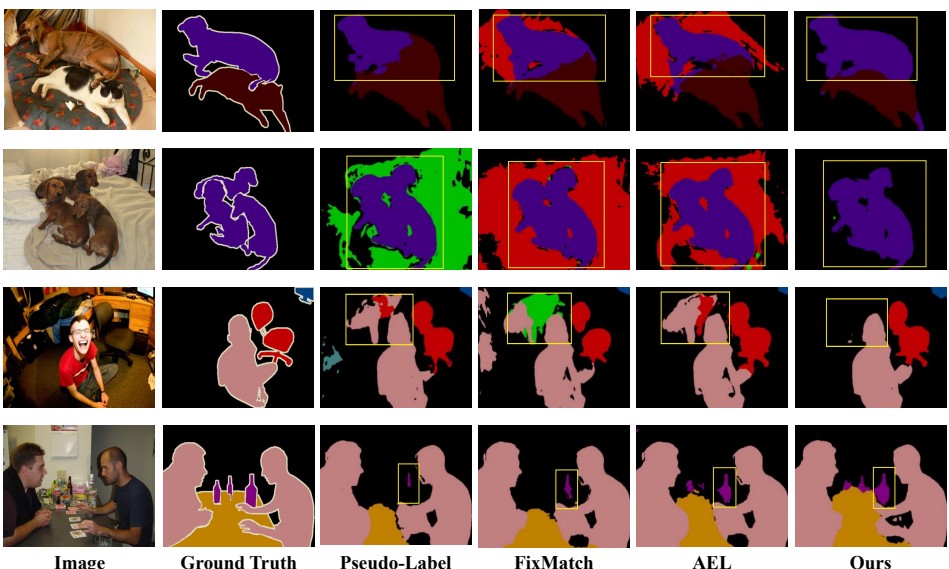

Figure 7: More qualitative comparison with different methods. Note that significant improvements are marked with yellow boxes.