# OpenReview forum: "DAW: Exploring the Better Weighting Function for Semi-supervised Semantic Segmentation"
_NeurIPS.cc/2023/Conference — NeurIPS 2023 poster_

### Official Review · Reviewer_UScw · 2023-06-25

**Soundness:** 4 excellent
**Presentation:** 3 good
**Contribution:** 4 excellent
**Rating:** 7
**Confidence:** 4

**Summary:**

This paper studies semi-supervised semantic segmentation, where the key challenge is how to select high-quality pseudo-labels. This paper systematically analyzes the trade-off between correct unreliable predictions and wrong confident predictions and theoretically finds the optimal weighting function. Interestingly, by explicitly modeling the confidence distributions of both correct and wrong predictions, the optimal weighting function is simply the hard step function with the jump point located at the intersection of the estimated two distributions. Experiments under various evaluation protocols demonstrate the efficacy of the proposed method.

**Strengths:**

1. This paper is well-written and very easy to follow.
2. Sufficient details are provided in the supplementary material.
3. The idea of minimizing $E_1+E_2$ is quite novel.
4. The proposed method surpasses state-of-the-art alternatives on various benchmarks, especially when the labeled images are extremely limited (e.g., 1/16 protocol on classic VOC).

**Weaknesses:**

There are some minor questions.
1. The detailed operation of the proposed Distribution Alignment is relatively confusing. Does "$\cdot$" in Eq. (12) indicate dot-product? It is better to provide a more detailed explanation of how Eq. (12) operates.
2. Will more precise estimations contribute to better performances? For instance, will re-computing $(\mu^{+}, \sigma^{+})$ and $(\mu^{-}, \sigma^{-})$ over the *whole* labeled set after each iteration bring improvements? Or, is there any other technique better than the simple EMA when estimating the parameters of distributions?
3. I am curious about the performance when applying DAW to other related fields, such as domain adaptive segmentation and semi-supervised classification/detection.

**Questions:**

I do not have any questions. Please refer to the weaknesses section.

**Limitations:**

The authors **did not** discuss limitations and broader impacts.

---

> ### Author Rebuttal · Authors · 2023-08-10
>
> Thanks for taking the time to share your comments and suggestions in the review assessment, as well as for acknowledging the good writing, sufficient supplementary material, novel idea and convincing experiment result.
> We see that your main concern is how to better estimate the distribution.
> All your comments are addressed point by point in the following.
>
> **Q1:** The detailed operation of the proposed Distribution Alignment is relatively confusing. Does "$\cdot$" in Eq. (12) indicate dot-product? It is better to provide a more detailed explanation of how Eq. (12) operates.
>
> **A1:** Sorry for not being able to give you an intuitive understanding in Eq. (12). The Distribution Alignment is proposed to narrow the discrepancy between the prediction distributions of labeled and unlabeled data to further unlock the potential of the Distribution-Aware Weighting Function.
> Specifically, we first calculate the class-wise prediction confidence expectation of the model for labeled data and unlabeled data.
> Then an element-wise division operation is done between the expectation of labeled data and unlabeled data, which represents the prediction discrepancy of the model w.r.t labeled data and unlabeled data.
> Finally, we rectify the prediction for unlabeled data in a modulation manner, i.e., "$\cdot$" denotes the element-wise multiplication operation.
> We will provide a more detailed explanation in the revised version to further improve the readability.
>
> **Q2:** Will more precise estimations contribute to better performances? For instance, will re-computing ($\mu^+, \sigma^+$) and ($\mu^-, \sigma^-$) over the whole labeled set after each iteration bring improvements? Or, is there any other technique better than the simple EMA when estimating the parameters of distributions?
>
> **A2:** The prerequisite for our method to work well is a good estimation of the inherent positive and negative distribution.
> Theoretically, re-computing the ($\mu, \sigma$) over the whole labeled set after each iteration leads to a more precise estimation while bringing the unacceptable cost of training.
> In fact, the model’s learning status changes not much between two adjacent iterations.
> Adopting EMA is an effective and efficient approach, which is proven by many current works such as FreeMatch [17], SoftMatch [23].
> Anyhow, exploring better estimation techniques is what we are committed to in the future.
>
> **Q3:** I am curious about the performance when applying DAW to other related fields, such as domain adaptive segmentation and semi-supervised classification/detection.
>
> **A3:** The mainstream approach to these tasks you enumerated also involves pseudo-label selection.
> And our method can theoretically minimize the negative equivalence effect brought by the pseudo-label.
> So we believe that our method can bring performance gains to these tasks.
> However, in this paper, we only consider the semi-supervised setting, where the labeled and unlabeled data come from the same domain, and  a simple Distribution Alignment can work well.
> While in the domain adaptation task, the labeled and unlabeled data come from different domains, meaning that the distribution discrepancy between the labeled and unlabeled data further increases.
> Only by effectively alleviating such a domain discrepancy can our DAWF play its greatest role.
> Thanks for your detailed suggestion, which is really worth thinking about in future works.
>
> We appreciate your positive review. Please let us know if you have further questions:)
>
> **Limitation and Society Impact.** Our method is primarily geared towards enhancing the utilization of unlabeled data, with comparatively lesser attention directed towards the labeled data.
> Exploring avenues to effectively harness the labeled data in semi-supervised semantic segmentation is also a promising investigation.
> Within this paper, we present an approach for semi-supervised semantic segmentation, a pivotal research area in the realm of computer vision, with no apparent negative societal implications known thus far.

---

> ### Comment · Reviewer_UScw · 2023-08-15
> **Post Rebuttal Comments by Reviewer UScw**
>
> I appreciate the rebuttal, which has addressed all my concerns. Therefore, I will raise my rating to 7.
>
> Also, I have some comments for Reviewer mWrq and SG4E.
> - **@Reviewer mWrq**, this paper presents a unified perspective. Not including some complicated methods is not a reason for rejection. Unifying a lot of previous common practices is the contribution.
> - **@Reviewer SG4E**, the proposed method does not need very precise estimation to achieve competitive results. I think this attribute has definitely demonstrated the effectiveness of the proposed unified perspective. A more precise estimation is expected to have better performance, but it seems to be no need for a perfect estimation.
>
> I am looking forward to an open dialog towards the contribution of this paper.

---

> > ### Author Response · Authors · 2023-08-16
> > **Thank you for acknowledgement of our response and score raising**
> >
> > Dear reviewer UScw,
> >
> > We sincerely appreciate your endorsement of our work and your positive feedback! We will make every effort to further improve our work!
> >
> > Authors

---

### Official Review · Reviewer_Sjij · 2023-07-01

**Soundness:** 3 good
**Presentation:** 3 good
**Contribution:** 3 good
**Rating:** 7
**Confidence:** 5

**Summary:**

In this paper, the authors thoroughly analyze the pixel-level pseudo label selection criteria and the trade-off associated with weighting functions in semi-supervised semantic segmentation. The authors propose a novel method called distribution-aware weighting (DAW) based on their analysis and confidence distribution modeling. The authors present sophisticated mathematical derivations and provide extensive experimental results to demonstrate the effectiveness of their approach. This research makes a valuable contribution to the field of semi-supervised semantic segmentation.

**Strengths:**

1. The authors establish a well-thought-out confidence modeling framework and conduct a comprehensive analysis of the trade-off involved in pseudo-labeling. The mathematical derivations presented in this paper are sophisticated and thoughtful. The theoretical analysis provides valuable insights and guidance for future research.
2. The theoretical analysis of the trade-off effectively demonstrates the effectiveness of the distribution-aware weighting (DAW) method and explains the reasons behind its superiority over competitor approaches. This mathematical tool not only validates the performance of DAW but also enables subsequent researchers to assess the theoretical performance of semi-supervised learning (SSL) techniques.
3. In addition to evaluating the effectiveness of this work on commonly used semi-supervised semantic segmentation benchmarks, the authors extend their analysis to include other benchmarks, such as Lucchi.


**Weaknesses:**

1.	The notation in L107, L108 and L109 are not well matched. The authors use N_L and N_U in L107, L108, but write this notation as N_l and N_u in the following.

**Questions:**

1. In the confidence distribution modeling approach proposed by the authors, the normalization factor (\alpha) for the truncated Gaussian distribution modeling the confidence distribution of pseudo-labels is not explicitly specified in the provided information. How is the value of the normalization factor obtained?
2. The specific implementation details regarding the semi-supervised semantic segmentation methods employed in the paper have not been mentioned. Which kind of semi-supervised semantic segmentation methods are used in your paper? Teacher-Student scheme? Or other kinds? Please point out detailed implementation.
3. The authors justify their choice of the Gaussian distribution for confidence distribution modeling based on the principle of maximum entropy and provide a detailed mathematical derivation to support their decision. However, it is worth considering if there are alternative explanations to support their opinion, such as experimental observations or more general assumptions.


**Limitations:**

The authors have not stated the limitations of their work.

---

> ### Author Rebuttal · Authors · 2023-08-10
>
> Thanks for taking the time to share your comments and suggestions in the review assessment, as well as for acknowledging the well-thought-out design, comprehensive analysis, sophisticated mathematical derivation and valuable insight.
> All your comments are addressed point by point in the following.
>
> **Q1:** The notation in L107, L108 and L109 are not well matched. The authors use $N_L$ and $N_U$ in L107, L108, but write this notation as $N_l$ and $N_u$ in the following.
>
> **A1:** Thank you for your patient reading and sorry for the clerical error. We mix up the $N_L/N_U$ and $N_l/N_u$ incorrectly. We will carefully check our manuscripts in the revised version.
>
> **Q2:** In the confidence distribution modeling approach proposed by the authors, the normalization factor ($\alpha$) for the truncated Gaussian distribution modeling the confidence distribution of pseudo-labels is not explicitly specified in the provided information. How is the value of the normalization factor obtained?
>
> **A2:** The effect of the normalization factor is to make the integral from $1/C$ to $1$ of the truncated Gaussian distribution 1 (this is the definition of the probability density function). We specify the value in L132 for $\alpha$ and L136 for $\beta$.
>
> **Q3:** The specific implementation details regarding the semi-supervised semantic segmentation methods employed in the paper have not been mentioned. Which kind of semi-supervised semantic segmentation methods are used in your paper? Teacher-Student scheme? Or other kinds? Please point out detailed implementation.
>
> **A3:** Following FixMatch[18], there is only one segmentation model in our method (Deeplabv3+), acting as teacher and student in the meanwhile.
> We will supply these descriptions in the revision.
>
> **Q4:** The authors justify their choice of the Gaussian distribution for confidence distribution modeling based on the principle of maximum entropy and provide a detailed mathematical derivation to support their decision. However, it is worth considering if there are alternative explanations to support their opinion, such as experimental observations or more general assumptions.
>
> **A4:** Maximum entropy means that we can make the assumption with minimal error expectations without any prior, and is the core property of the Gaussian distribution, which is why the Gaussian distribution is so widely used, even in the Batchnorm.
>
> We appreciate your positive review. Please let us know if you have further questions:)
>
> **Limitation and Society Impact.** Our method is primarily geared towards enhancing the utilization of unlabeled data, with comparatively lesser attention directed towards the labeled data.
> Exploring avenues to effectively harness the labeled data in semi-supervised semantic segmentation is also a promising investigation.
> Within this paper, we present an approach for semi-supervised semantic segmentation, a pivotal research area in the realm of computer vision, with no apparent negative societal implications known thus far.

---

> > ### Comment · Reviewer_Sjij · 2023-08-20
> >
> > Thanks for the response from the authors. I will keep my previous rating.

---

> ### Comment · Area_Chair_H3Qd · 2023-08-19
>
> This is a friendly reminder from the AC that you need to respond to the rebuttal, since the authors spent quite a lot of time preparing the rebuttal.

---

### Official Review · Reviewer_SG4E · 2023-07-04

**Soundness:** 3 good
**Presentation:** 3 good
**Contribution:** 2 fair
**Rating:** 4
**Confidence:** 4

**Summary:**

This paper discusses the challenge of utilizing unlabeled data to improve the generalization performance of semantic segmentation models. Existing methods use weighting functions to select pseudo-labels, but face a trade-off between accuracy and utilization. The proposed method, DAW, models the confidence distribution of pseudo-labels and uses a unified weighting function to define the trade-off problem. The authors find the optimal solution to be a hard step function and propose distribution alignment to mitigate prediction distribution discrepancies.

**Strengths:**

1. A new method called DAW is proposed, which models the confidence distribution of correct and inaccurate pseudo-labels and uses a unified weighting function to define the trade-off problem. This method achieves better results in handling pseudo-labels compared to existing methods.
2. The hard step function and distribution alignment are used to solve the trade-off problem and prediction distribution discrepancy problem.
3. The writing and presentation of the paper are well-done.

**Weaknesses:**

1. The premise for the proposed method to work is to correctly estimate the distribution of correct and incorrect pseudo-labels, which is a difficult problem. Firstly, for unlabeled data, it is very challenging to distinguish which pseudo-labels are correct and which are incorrect because there is no ground truth. Secondly, estimating the distribution based on labeled data may not guarantee the effectiveness of all data because in most scenarios, the proportion of labeled data is low.
2. The paper uses a hard step function as the weighting function. What are the advantages of using a hard step function over a regular step function with a threshold set as a hyperparameter?
3. From the results of Table 2, 3, it can be seen that the improvement in the final performance in most scenarios is relatively minimal compared to previous works.

**Questions:**

What kind of results would we get if we combine the DA method with a manual approach of setting the most appropriate hyperparameters while using the FixMatch method?

**Limitations:**

The paper does not illustrate any limitations or any potential negative societal impact.

---

> ### Author Rebuttal · Authors · 2023-08-10
>
> Thanks for taking the time to share your comments and suggestions in the review assessment, as well as for acknowledging the novelty and well-done presentation.
> We see that your main concern is about the effectiveness of DAW.
> All your comments are addressed point by point in the following.
>
> **Q1:** The premise for the proposed method to work is to correctly estimate the distribution of correct and incorrect
> pseudo-labels, which is a difficult problem.
> Firstly, for unlabeled data, it is very challenging to distinguish which pseudo-labels are correct and which are incorrect because there is no ground truth.
> Secondly, estimating the distribution based on labeled data may not guarantee the effectiveness of all data because in most scenarios, the proportion of labeled data is low.
>
> **A1:** We agree with the statement that "The premise for ..., which is a difficult problem".
> However, this is the exact issue we are striving to alleviate in DAW.
>
> To distinguish which pseudo-labels are correct and which are incorrect, unlike previous methods such as FixMatch and AEL that fail to utilize the unlabeled data more effectively since they either use a pre-defined fixed threshold or an ad-hoc threshold adjusting scheme,
> compromising their capability.
> We formally define the trade-off between inaccurate yet utilized pseudo-labels, and correct yet discarded pseudo-labels by explicitly modeling the confidence distribution of correct and inaccurate pseudo-label, and derive the better weighting function (hard step function with the jump point located at the intersection of the two confidence distributions), minimizing the negative equivalence impact raised by the trade-off, that is, *maximizing the separation of correct and incorrect pseudo-labels*.
>
> Furthermore, considering the discrepancy between the prediction distributions of labeled and unlabeled data, we devise distribution alignment to *narrow the gap and enhance applicability to all data*, in pursuit of further unlocking the potential of the weighting function and enjoying the synergy (Please refer to Figure 4 in the main paper).
>
> Extensive experiments on mainstream benchmarks including electron microscope images demonstrate the effectiveness of DAW, especially when the labeled images are extremely limited (+6.3% gains under 1/16 partition protocol with ResNet-50 on VOC classic set compared to recent SOTA  AugSeg in Table 2).
>
> Note that similar paradigms have been proven effective, where the prediction distribution of labeled data is utilized to fit the unlabeled data.
> For example, AEL employs labeled data to estimate class-wise confidence and guide the sampling and re-weighting of unlabeled data. We will add these discussions in the revised version for better clarification.
>
> **Q2:** The paper uses a hard step function as the weighting function. What are the advantages of using a hard step function over a regular step function with a threshold set as a hyperparameter?
>
> **A2:**
> In fact, we do NOT take the hard step function as a prerequisite. Instead, we formally define the trade-off by explicitly modeling the confidence distribution of correct and inaccurate pseudo-labels, and derive the optimal solution for the weighting function is a hard step function (referred to as distribution-aware weighting function, DAWF), with the jump point located at the intersection of the two confidence distributions.
> Note that the dedicated weighting function is theoretically guaranteed, and is free of setting thresholds manually compared to the regular step function with a predefined fixed threshold.
>
> Besides, due to the negative/positive distribution evolving with the model's learning status, the DAWF dynamically adjusts the threshold for maximizing the separation between correct and inaccurate pseudo-labels (see Figure 5 in the main paper).
> This is distinct from regular step functions that are arbitrarily controlled by hyper-parameters and thus disconnected from the model’s learning process, resulting in inferior performance.
>
> Thanks for the suggestion. We further combine Distribution Alignment (DA) with regular step function using different hyperparameters to validate the effectiveness of DAWF. As tabulated in the following Table.
> |         ResNet-50         | Ours (DA + DAWF) | DA + 0.95 | DA + 0.9 | DA + 0.8 | DA + 0.6 | DA + 0 | FixMatch |
> | :-----------------------: | :--------------: | :-------: | :------: | :------: | :------: | :----: | :----------: |
> | Pascal blender 1/16 (662) |       75.8       |   71.0    |   70.2   |   69.7   |   69.8   |  69.5  |     70.6     |
>
> We have the following additional findings: (1) The introduction of DAWF exceeds all fixed threshold settings under the 1/16 partition protocol, which verifies the effectiveness of DAWF.
> (2) DA is devised to narrow the gap between the prediction distributions of labeled and unlabeled data, in pursuit of further unlocking the potential of the weighting function and enjoying the synergy. Therefore, From the 2nd and 8th columns, we can observe that performance gains from simply adopting the DA on the FixMatch with threshold=0.95 are negligible.
> (3) We observe that entries with low thresholds performed worse than those with high thresholds. We conjecture that this is due to the fact that while low threshold hyperparameters increase the utilization of pseudo-labels, they inevitably recruit noisy pseudo labels into training, leading to confirmation bias, which is a corollary raised by erroneous pseudo-labels.
>
> Please refer to the **global response** for Rebuttal -- Part 2.

---

> ### Comment · Area_Chair_H3Qd · 2023-08-19
>
> This is a friendly reminder from the AC that you need to respond to the rebuttal, since the authors spent quite a lot of time preparing the rebuttal.

---

### Official Review · Reviewer_mWrq · 2023-07-06

**Soundness:** 2 fair
**Presentation:** 2 fair
**Contribution:** 2 fair
**Rating:** 5
**Confidence:** 4

**Summary:**

This paper presents a systematic analysis of the trade-off involved in previous methods that deal with pseudo-labels. The authors define the trade-off between inaccurate yet utilized pseudo-labels and correct yet discarded pseudo-labels by modeling the confidence distribution of both types of pseudo-labels. They propose a unified weighting function called DAW to minimize the negative impact caused by this trade-off. Interestingly, they find that the optimal solution for the weighting function is a hard step function, located at the intersection of the confidence distributions. Additionally, the authors introduce distribution alignment to address the discrepancy between the prediction distributions of labeled and unlabeled data. The experimental results on various benchmarks, including mitochondria segmentation, demonstrate that DAW outperforms state-of-the-art methods.

**Strengths:**

+ The paper is presented in a clear and concise manner, effectively communicating the motivation and ideas behind the proposed approach. The authors thoroughly analyze the trade-off involved in handling pseudo-labels in previous methods, defining it as the balance between utilizing inaccurate pseudo-labels and discarding correct ones.
+ The paper provides a rigorous evaluation of the proposed approach through extensive experiments conducted on multiple benchmarks, ensuring a comprehensive assessment of its performance. The results obtained from the experiments showcase the superiority of the proposed approach against state-of-the-art methods.

**Weaknesses:**

- It is not accurate to state that existingmethods tend to employ certain criteria (weighting function) to select pixel-level pseudo labels. For instance, FlexMatch [40] dynamically adjusts thresholds for different classes at each time step to allow informative unlabeled data and their corresponding pseudo labels to be included, and FreeMatch [17] employs a self-adaptive approach to adjust the confidence threshold based on the learning status of the model.
- The current state-of-the-art method [a] is not included for comparison in Tables 2-3.
[a] "NP-SemiSeg: When Neural Processes meet Semi-Supervised Semantic Segmentation." ICML. 2023.

**Questions:**

Please refer to the Weaknesses.

**Limitations:**

The authors have adequately addressed the limitations

---

> ### Author Rebuttal · Authors · 2023-08-10
>
> Thanks for taking the time to share your comments and suggestions in the review assessment, as well as for acknowledging the thorough analysis, rigorous experiments, and well-structured presentation.
> We see that your main concern is about the imprecise statement.
> All your comments are addressed point by point in the following.
>
> **Q1:** It is not accurate to state that existing methods tend to employ certain criteria (weighting function) to select pixel-level pseudo labels. For instance, FlexMatch [40] dynamically adjusts thresholds for different classes at each time step to allow informative unlabeled data and their corresponding pseudo labels to be included, and FreeMatch [17] employs a self-adaptive approach to adjust the confidence threshold based on the learning status of the model.
>
> **A1:** Sorry for any misunderstanding.
> Our intention is not to underscore that we are the first work to design a threshold adjustment strategy conditioned on the model’s learning status.
> Instead, we just want to summarize how mainstream methods in *semi-supervised semantic segmentation task* handle pseudo-labels (although, to our knowledge, most methods employ predefined criteria without considering the model's learning status).
>
> For the mentioned semi-supervised learning methods including FlexMatch and FreeMatch, we argue that they might be insufficient in terms of adjusting thresholds according to the model’s learning progress ascribed to the lack of sophisticated consideration, thus impeding the training process.
> To be more specific, FlexMatch dynamically adjusts class-wise thresholds to seek model status awareness, but these class-specific thresholds are still mapped from a *predefined fixed global threshold* (e.g., 0.95).
> Furthermore, FreeMatch attempts to empower the global threshold to perceive the learning status, but it is still constrained by the ad hoc definition of the average prediction confidence as a replacement for the previously fixed threshold.
>
> Orthogonal to these methods, our DAW  formally defines the trade-off between inaccurate yet utilized pseudo-labels, and correct yet discarded pseudo-labels by explicitly modeling the confidence distribution of correct and inaccurate pseudo-labels, and derives the better weighting function (i.e., pseudo-label selection criteria), minimizing the negative equivalence impact raised by the trade-off.
> To further validate this point, we re-implement these two methods and compared them with our DAW  on the VOC blender set under 1/16 partition protocol, as summarized in the following Table, confirming the effectiveness of DAW.
> |   ResNet-50   | Pascal Blender 1/16 (662) |
> | :-----------: | :-----------------------: |
> | FlexMatch |           72.9            |
> | FreeMatch |           74.1            |
> |     Ours      |           75.8            |
>
> We will improve the writing of this part and add these discussions in the revised paper.
>
>
> **Q2:** The current state-of-the-art method NP-SemiSeg is not included for comparison in Tables 2-3.
>
> **A2:** NP-Semiseg is a very recent work released later than the NeurIPS paper submission.
> But thanks for mentioning this related work, we add its comparison with DAW on VOC classic set and will include the full comparison and discussion in the revised paper.
> | Pascal classic ResNet-50 | 1/16 (92) | 1/8 (183) | 1/4 (366) | 1/2 (732) |
> | :----------------------: | :-------: | :-------: | :-------: | :-------: |
> |      NP-SemiSeg      |   65.78   |   72.4    |   75.8    |   77.4    |
> |           Ours           |   71.5    |   72.4    |   76.7    |   78.1    |
>
> We hope our response can resolve your concern on DAW. Please let us know if you still have any concern regarding the paper:)

---

> ### Comment · Area_Chair_H3Qd · 2023-08-19
>
> This is a friendly reminder from the AC that you need to respond to the rebuttal, since the authors spent quite a lot of time preparing the rebuttal.

---

> ### Author Response · Authors · 2023-08-19
> **Sincerely Look Forward to Your Feedback**
>
> Dear Reviewer mWrq,
>
> Thanks again for your insightful suggestions and comments. As the deadline for reviewer-author discussion is approaching, we are glad to provide any additional clarifications that you may need.
>
> We have carefully studied your comments and added additional clarifications and analysis in our previous responses to address your concerns to the best of our ability. We genuinely hope you could kindly check our response.
>
> We hope that our previous responses have convinced you of the merits of our work. If our response addressed your concern, please consider raising the score. Please do not hesitate to contact us if there are other clarifications or experiments we can offer.
>
> Thank you for your time again.
>
> Best wishes,
> Authors

---

### Official Review · Reviewer_ahfa · 2023-07-08

**Soundness:** 3 good
**Presentation:** 2 fair
**Contribution:** 2 fair
**Rating:** 5
**Confidence:** 1

**Summary:**

This paper introduces DAW, a method for semi-supervised semantic segmentation that addresses the trade-off between false-positive and false-negative pseudo-labels. DAW utilizes Gaussian functions to model confidence distributions and derives an optimal weighting function based on their inter-section. It also incorporates distribution alignment to reduce the discrepancy between labeled and unlabeled data. Experimental results demonstrate the superiority of DAW over state-of-the-art methods on two benchmarks.

**Strengths:**

- The author's viewpoint offers valuable insights into the problem of semi-supervised semantic segmentation.
- The technical approach proposed in the paper is described adequately.

**Weaknesses:**

- While the paper demonstrates the efficacy of DAW on certain benchmarks, it would be beneficial to investigate the method's performance on larger datasets such as COCO-Stuff, ADE20K, and COCO to provide a comprehensive evaluation across different scales.
- Table 1 is challenging to interpret without additional illustration of the symbols used in both the table and figure. Further clarification and explanation are needed to enhance readability.
- The English writing in the paper requires further improvement. For example:
    + In line 1, revise "lies how to fully" to "lies in how to fully."
    + In line 2, rephrase "inaccurate yet utilized pseudo-labels, and correct yet discarded pseudo-labels" to "false positive pseudo-labels" and "false negative pseudo-labels" respectively.

---
Post rebuttal:
After carefully reviewing the authors' feedback, I have made the decision to maintain my rating, which falls within the realm of borderline acceptance.

**Questions:**

Please refer to the Weaknesses.

**Limitations:**

Yes.

---

> ### Author Rebuttal · Authors · 2023-08-10
>
> Thanks for taking the time to share your comments and suggestions in the review assessment, as well as for acknowledging the novelty and adequate technical approach.
> We see that your main concerns are more experiments on a larger dataset and further explanation in Table 1.
> All your comments are addressed point by point in the following.
>
> **Q1:** While the paper demonstrates the efficacy of DAW on certain benchmarks, it would be beneficial to investigate the method's performance on larger datasets such as COCO-Stuff, ADE20K, and COCO to provide a comprehensive evaluation across different scales.
>
> **A1:** Thanks for this valuable suggestion, to comprehensively assess the performance of our DAW against other competitors, we conduct experiments on the COCO dataset, which comprises 118k/5k training/validation images with 81 classes and can serve as quite a challenging benchmark.
>
> For a fair comparison, we adopt exactly the same setting following PseudoSeg [1] and PC$^2$Seg [2].
> As tabulated in Table, we can observe that DAW achieves consistent performance gains over the Sup.-only baseline, obtaining improvements of 11.6%, 10%, and 7.8% under 1/128, 1/64, and 1/32 partition protocols, respectively.
> Besides, DAW also has a clear lead of ~4% compared to the recent SOTA method (PC$^2$Seg), which confirms the effectiveness of DAW to minimize the negative equivalence impact raised by the trade-off between inaccurate yet utilized pseudo-labels, and correct yet discarded pseudo-labels in previous methods.
> |     COCO      | 1/128 (925) | 1/64 (1849) | 1/32 (3697) |
> | :-----------: | :---------: | :---------: | :---------: |
> |   Sup.-only   |    33.6     |    37.8     |    42.2     |
> | PseudoSeg [1] |    39.1     |    41.8     |    43.6     |
> | PC$^2$Seg [2] |    40.1     |    43.7     |    46.1     |
> |     Ours      |    45.2     |    47.8     |    50.0     |
> We will supply the related results in the revision.
>
> **Q2:** Table 1 is challenging to interpret without additional illustrations of the symbols used in both the table and figure. Further clarification and explanation are needed to enhance readability.
>
> **A2:** Sorry for not being able to give you an intuitive understanding of Table 1.
> We have added additional symbol explanations to help readers better understand the trade-off in previous methods that hinder the model’s learning when dealing with pseudo-labels in semi-supervised semantic segmentation.
> Please refer to the attached PDF in the **global response** for more details.
> We will improve the writing of this part.
>
> **Q3:** The English writing in the paper requires further improvement.
>
> **A3:** We sincerely appreciate your patience during the review, and will further improve the writing in the revised version.
>
> Hope our response could resolve your concern. Please let us know if you have further questions:)
>
> [1] Pseudoseg: Designing pseudo labels for semantic segmentation. ICLR 2021.
>
> [2] Pixel contrastive-consistent semisupervised semantic segmentation. ICCV 2021.

---

> > ### Author Response · Authors · 2023-08-21
> > **Sincerely Look Forward to Your Feedback**
> >
> > Dear Reviewer ahfa,
> >
> > Thanks again for your insightful suggestions and comments. As the deadline for reviewer-author discussion is approaching. We are glad to provide any additional clarifications that you may need.
> >
> > We have carefully studied your comments and added additional clarifications and experiments in our previous responses to address your concerns. We genuinely hope you could kindly check our response.
> >
> > We hope that our previous responses have convinced you the merits of our work. Please do not hesitate to contact us if there are other clarifications or experiments we can offer.
> >
> > Thank you for your time again.
> >
> > Best wishes,
> >
> > Authors

---

> ### Comment · Area_Chair_H3Qd · 2023-08-19
>
> This is a friendly reminder from the AC that you need to respond to the rebuttal, since the authors spent quite a lot of time preparing the rebuttal.

---

### Author Rebuttal · Authors · 2023-08-10

**To Reviewer SG4E -- Rebuttal Part2:**

**Q3:** From the results of Tables 2 and 3, it can be seen that the improvement in the final performance in most scenarios is relatively minimal compared to previous works.

**A3:**
Semi-supervised semantic segmentation is a very challenging task, and DAW indeed provides some significant and promising improvement especially when the labeled images are extremely limited as recognized by the other four reviewers (+4.6% gains under 1/16 partition protocol with ResNet-101 on VOC classic set compared to recent SOTA AugSeg in Table 2).

On the other hand, since the purpose of this paper is to explore the better weighting function to alleviate the negative impact raised by the trade-off when dealing with pseudo-labels, we adopt FixMatch as a baseline following the simplicity principle.
In fact, the most direct and effective way to improve performance is data augmentation (e.g., AugSeg), which is also a mainstream direction of semi-supervised learning technologies.
Theoretically, the techniques can also be incorporated into our method to optimize the overall performance.
To validate our points, we combine data augmentation strategies from AugSeg with the pseudo-label selection criterion from our DAW, observing further gains in overall performance on the VOC dataset.
|   ResNet-50   | Pascal blender 1/16 (662) |
| :-----------: | :-----------------------: |
|  AugSeg   |           74.7            |
|     Ours      |           75.8            |
| Ours + AugSeg |           76.9            |

**Q4:** What kind of results would we get if we combine the DA method with a manual approach of setting the most
appropriate hyperparameters while using the FixMatch method?

**A4:** Thanks for the suggestion. In the **A2**, we address all your questions about the effectiveness of DAWF. In addition, we will add all these valuable discussions into the revised version or the supplementary material.


We hope our response can resolve your concern. Please do not hesitate to let us know if you have further questions:)

**Limitation and Society Impact.** Our method is primarily geared towards enhancing the utilization of unlabeled data, with comparatively lesser attention directed towards the labeled data.
Exploring avenues to effectively harness the labeled data in semi-supervised semantic segmentation is also a promising investigation.
Within this paper, we present an approach for semi-supervised semantic segmentation, a pivotal research area in the realm of computer vision, with no apparent negative societal implications known thus far.

---

### Decision · Program_Chairs · 2023-09-21

**Decision:**

Accept (poster)

**Comment:**

After rebuttal, four reviewers land on the positive side and only one reviewer (SG4E) leans to reject. Specifically, Reviewer SG4E finally leaves the initial rating – “Borderline Reject” unchanged and state that the authors solved some of his major problems in the rebuttal. However, neither which major problems are unsolved are clarified nor the justifications for the final rating are provided. Considering that other reviewers all enjoy the insights offered by the theoretical analysis and their major concerns, e.g., the lack of comparison on larger datasets such as COCO, have been addressed in the rebuttal, the AC decides to accept the paper. The authors should include the discussions and new results in the rebuttal into the final version.